# Extracellular vesicle-derived circCEBPZOS attenuates postmyocardial infarction remodeling by promoting angiogenesis via the miR-1178-3p/ PDPK1 axis

Ling Yu[1,2,6], Yubin Liang[1,6], Minzhou Zhang [1,2,3✉], Phillip C. Yang[4], Aleksander Hinek[5] & Shuai Mao [1,2,3,4✉]

Emerging studies indicate that extracellular vesicles (EVs) and their inner circular RNAs (circRNAs), play key roles in the gene regulatory network and cardiovascular repair. However, our understanding of EV-derived circRNAs in cardiac remodeling after myocardial infarction (MI) remains limited. Here we show that the level of circCEBPZOS is down-regulated in serum EVs of patients with the adverse cardiac remodeling compared with those without post-MI remodeling or normal subjects. Loss-of-function approaches in vitro establish that circCEBPZOS robustly promote angiogenesis. Overexpression of circCEBPZOS in mice attenuates MI-induced left ventricular dysfunction, accompanied by a larger functional capillary network at the border zone. Further exploration of the downstream target gene indicates that circCEBPZOS acts as a competing endogenous RNA by directly binding to miR-1178-3p and thereby inducing transcription of its target gene phosphoinositide-dependent kinase-1 (PDPK1). Together, our results reveal that circCEBPZOS attenuates detrimental post-MI remodeling via the miR-1178-3p/PDPK1 axis, which facilitates revascularization, ultimately improving the cardiac function.

[1] The Second Clinical College, Guangzhou University of Chinese Medicine, Guangzhou 510405, China. [2] Department of Critical Care Medicine, Guangdong Provincial Hospital of Chinese Medicine, Guangzhou 510120, China. [3] Guangdong Provincial Branch of National Clinical Research Centre for Chinese Medicine Cardiology, Guangzhou 510120, China. [4] Department of Medicine, Division of Cardiovascular Medicine, Stanford University School of Medicine, Stanford, CA 94305, USA. [5] Translational Medicine, Hospital for Sick Children, Toronto M5G 0A4, Canada. [6]These authors contributed equally: Ling Yu, Yubin Liang. ✉email: minzhouzhang@gzucm.edu.cn; maoshuai@gzucm.edu.cn

Although pharmacological and technological treatment strategies have been established, progression to heart failure occurs in up to one-third of myocardial infarction (MI) patients because of adverse cardiac remodeling[1]. Pathological left ventricular (LV) remodeling after MI is one of the most important risk factors for the development of complicated ventricular arrhythmia, congestive heart failure and even ultimate cardiogenic death[2]. Meaningfully, amplified myocardial angiogenesis that facilitates neovascularization is mainly responsible for structural preservation of infarcted myocardium and maintaining its basic function[3]. Therefore, the promotion of myocardial angiogenesis has been proposed as an efficient therapeutic approach for cardiac dysfunction and deleterious remodeling after MI.

Recent studies have focused on this phenomenon and explored whether extracellular vesicles (EVs), membranous vesicular bodies typically 30–200 nm in diameter, might be also involved in its regulation[4]. EVs contain multiple proteins, mRNAs and noncoding RNAs, contributing to cellular communication and the regulation of multiple processes. The number and content of heterogeneous EVs not only affect the physiological state of different normal cells but can also be linked to many pathological pathways[5]. Importantly, it has already been demonstrated that EVs play crucial roles in the development of cardiovascular diseases, specifically, in the regulation of post-MI cardiac remodeling[6]. Thus, it has been shown that CD4-activated EVs promote postischemic cardiac fibrosis through miR-142-3p/Wnt signaling cascade-mediated activation of interstitial myofibroblasts[7]. This finding suggested that pharmacologic targeting of miR-142-3p in CD4-activated EVs may hold promise for alleviation of post-MI cardiac remodeling. Moreover, the application of EVs isolated from the plasma of ischemia-conditioned rats improved cardiac function and angiogenesis after MI by targeting the 70-kDa heat shock protein[8]. Song et al., also showed that the application of EVs derived from umbilical blood mesenchymal stem cells attenuated myocardial injury by inhibiting ferroptosis in mice with experimental MI[9]. In addition, the use of certain EV-derived noncoding RNAs, such as the miRNAs and lncRNAs has been proven beneficial for healing of the human post-MI myocardium. For example, EV-derived lncRNA AK139128 derived from hypoxic cardiomyocytes promoted apoptosis and inhibited the proliferation of cardiac fibroblasts[10]. LncRNA KLF3-AS1 isolated from human mesenchymal stem cell-derived EVs ameliorates pyroptosis of cardiomyocytes and alleviates the outcome of MI through the induction of the miR-138-5p/sirtuin1 axis[11]. Moreover, EVs isolated from coronary serum of patients with MI promote myocardial angiogenesis through the miRNA-143/insulin-like growth factor-I receptor pathway[12]. Together, the abovementioned results indicate that noncoding RNAs may indeed play crucial roles in the regulation of different processes contributing to the final clinical outcome of patients afflicted with MI.

More precisely, the circular noncoding RNAs (circRNAs), which are widely present in eukaryotic cells and participate in the regulation of the transcription and posttranscriptional expression of multiple genes contributing to normal cardiac functions, might also regulate the development of certain cardiac pathologies. Meaningfully, it has been proposed that certain changes in circRNA levels might be considered potential biomarkers of the therapeutic efficiency of cardiovascular diseases[13]. Indeed, circRNA Hipk3 was demonstrated to contribute to cardiac regeneration after experimental MI in mice by binding to Notch1 and miR-133a[14]. Elevation of circRNA 010567 was also associated with the improvement of cardiac function and alleviation of myocardial fibrosis in MI rats by inhibiting transforming growth factor (TGF)- β1[15]. Moreover, circRNA Ttc3 in cardiomyocytes counteracted the hypoxia-induced ATP depletion and the deterioration of cardiac dysfunction by sponging miR-15b in a rat model of MI[16]. Exsomal circRNA 0001273 derived from human umbilical cord mesenchymal stem cells remarkably inhibited the occurrence of myocardial cell apoptosis and subsequently promoted MI repair in an ischemic environment[17]. However, other putative regulatory mechanisms by which EV-derived circRNAs contribute to postinfarct cardiac remodeling remain to be better explored.

To provide a basis for further study of the molecular pathogenesis of LV remodeling after MI and identify potential therapeutic targets for this disease, we compared the expression profiles of EV-derived circRNAs in patients with and without cardiac remodeling using high-throughput RNA sequencing. First, we tested the function of EVs-derived circCEBPZOS, which has already been related to LV remodeling, in vitro and in vivo using silencing and overexpression strategies. Then, we explored the mechanisms underlying the actions of circCEBPZOS during the LV remodeling after MI. We anticipated that the obtained results would improve our understanding of EV-derived circCEBPZOS regulation during the development of postinfarct cardiac remodeling.

## Results

**EVs isolated from the serum of patients with post-MI cardiac remodeling contain significantly downregulated levels of circCEBPZOS.** Our initial whole transcriptome sequencing analysis aimed to explore circRNA expression in EVs derived from the sera of patients with and without postinfarct cardiac remodeling. The RNA sequencing data have been deposited in NCBI's Gene Expression Omnibus and are accessible through GEO Series accession number GSE194388. The obtained results indicated that three differentially expressed EV-derived circRNAs (hsa_circ_0000212, hsa_circ_0089282 and circCEBPZOS) could be detected in the comparison between the postinfarct cardiac remodeling (CR) and control groups and between the CR and non-postinfarct cardiac remodeling (N-CR) groups (Fig. 1a).

Importantly, we found that while the expression of hsa_circ_0000212 and hsa_circ_0089282 was significantly higher, the expression of circCEBPZOS was downregulated in both the CR vs. N-CR group, and the CR vs. N-CR group (Fig. 1b, c). To confirm these results, qRT–PCR was performed to detect the expression of these circRNAs by using divergent primers in the serum EVs of patients with and without post-MI cardiac remodeling. The isolated EVs were identified by transmission electron microscope (TEM) and nanoparticle tracking analysis (NTA). The results showed that the diameter of EVs was between 100–150 nm, which presented the typical EV' morphology (Fig. 1d). Then, we additionally found that the EV biomarkers, CD6, CD9 and TSG101 were also highly detected in all groups (Fig. 1e). Consistent with the transcriptome analysis, the qRT–PCR results showed that the expression of hsa_circ_0000212 and hsa_circ_0089282 was significantly elevated while the expression of circCEBPZOS was downregulated in the comparison between both the CR vs. N-CR groups and in the CR vs. control groups (Fig. 1f).

**Characteristics of circCEBPZOS.** To learn whether circCEBPZOS functions as a circRNA during postinfarct cardiac remodeling, first, the ring structure of circCEBPZOS was detected (Fig. 2a). The specific products of circCEBPZOS were probed with the divergent and convergent primers. The agarose electrophoresis assay demonstrated that circCEBPZOS could be amplified from both cDNA and gDNA templates by using convergent primers, while it could only be amplified from the cDNA template by using divergent primers (Fig. 2b). However, the

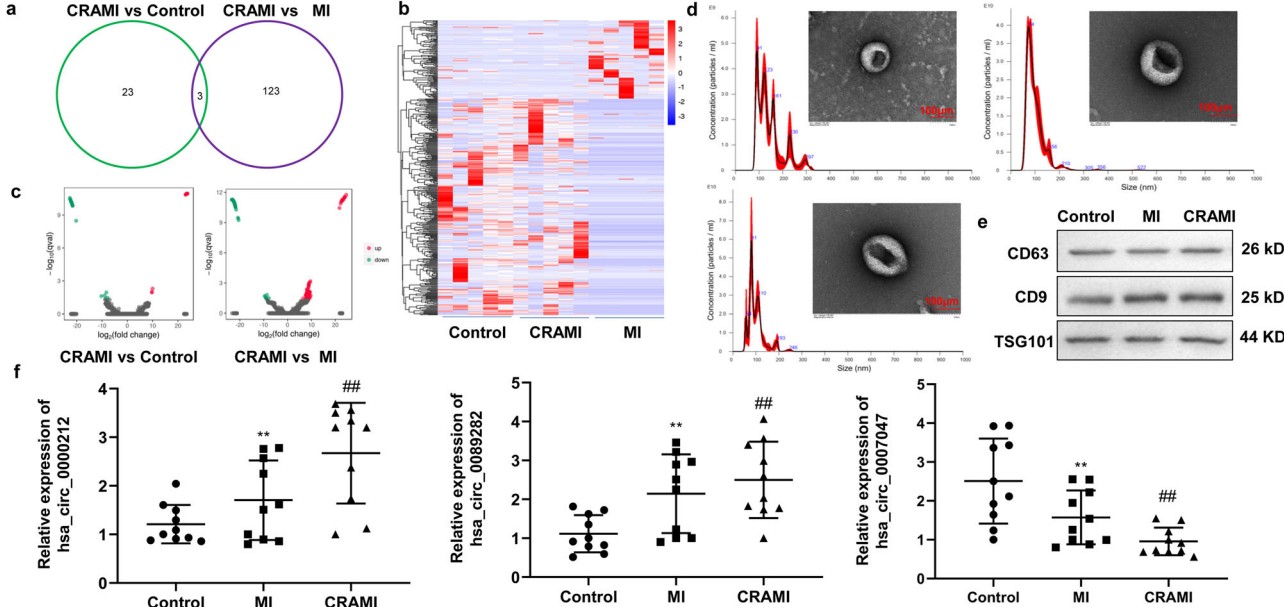

**Fig. 1 circCEBPZOS was significantly downregulated in the serum EVs of patients with postinfarct cardiac remodeling. a** Venn diagram to present the number of differentially expressed circRNAs among the serum exosomes from patients with and without postinfarct cardiac remodeling using whole transcriptome sequencing. **b**, **c** Heatmap and volcano plot to present the expression of differentially expressed circRNAs, respectively. **d** Phenotype of exosomes isolated from serum from patients with and without postinfarct cardiac remodeling by transmission electron microscopy, Bar = 100 μm; the particle size of exosomes identified by nanoparticle tracking analysis. **e** Protein expression of CD6, CD9 and TSG101 was detected using western blotting. **f** Expression of hsa_circ_0000212, hsa_circ_0089282 and circCEBPZOS in exosomes from patients with and without postinfarct cardiac remodeling by using divergent primers. GAPDH served as control. Data are mean ± SD. **P < 0.01, N-CR group vs. Control group; ##P < 0.01, CR group vs N-CR group. CR postinfarct cardiac remodeling, N-CR nonpostinfarct cardiac remodeling.

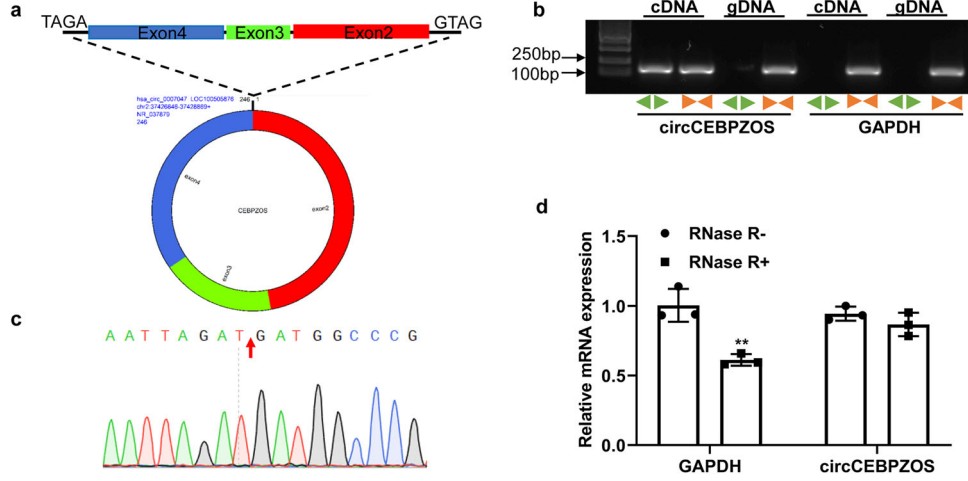

**Fig. 2 Characteristics of circCEBPZOS. a** Schematic diagram of the ring structure of circCEBPZOS. **b** Agarose electrophoresis assay to amplify the specific products of circCEBPZOS from both cDNA and gDNA templates by using convergent and divergent primers. GAPDH served as a control. **c** Reverse shear site of circCEBPZOS was confirmed by Sanger sequencing of the amplified product of circCEBPZOS; **d** qRT–PCR was used to detect the expression of circCEBPZOS by using cDNA as a templates with the RNA digested by RNA exonuclease R before reverse transcription. GAPDH served as the negative control. Data are mean ± SD. **P < 0.01, RNase R + group vs. RNase R- group.

GAPDH could only be amplified by the convergent primers, using cDNA and gDNA as the template.

Moreover, the reverse shear site of circCEBPZOS was confirmed by Sanger sequencing of the amplified product of circCEBPZOS (Fig. 2c). Finally, qRT–PCR was performed with cDNA as templates after the RNA was digested by RNA exonuclease R before reverse transcription, and GAPDH was used as the negative control. The results showed that the expression of circCEBPZOS was not changed in RNase R( + ) group compared to the RNase R(-) group. The expression of

GAPDH was significantly downregulated in the RNase R( + ) group compared to the RNase R(-) group, indicating that circCEBPZOS had a relatively stable structure (Fig. 2d).

**circCEBPZOS promotes angiogenesis, migration and proliferation of cultured VSMCs.** Angiogenesis has been shown to play a crucial role in postinfarct cardiac remodeling[18]. Therefore, we explored whether circCEBPZOS affects angiogenesis of CMECs and the proliferation and migration of human vascular smooth muscle cells (VSMCs). We found that circCEBPZOS was

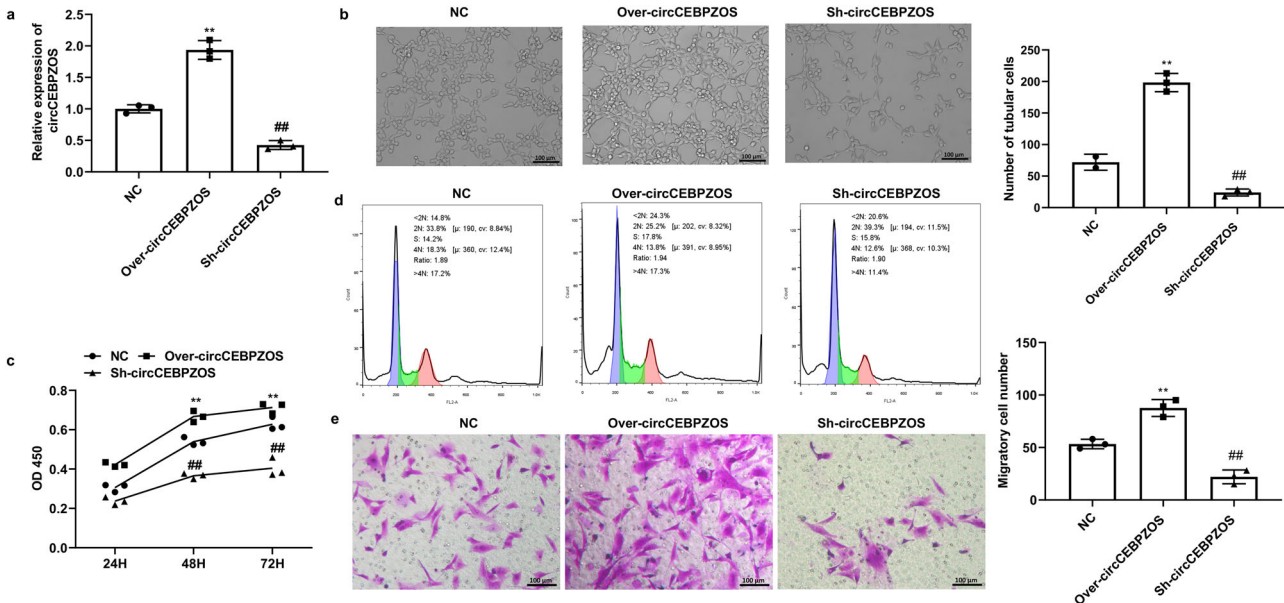

**Fig. 3 circCEBPZOS promotes angiogenesis of CMECs and proliferation and migration of VSMCs. a** qRT–PCR analysis was used to detect the expression of circCEBPZOS in the coincubation with EVs harboring with Over-circCEBPZOS or Sh-circCEBPZOS, GAPDH served as control; **b** Capillary-like structure analysis by tube formation assay in the Over-circCEBPZOS or Sh-circCEBPZOS group compared with the NC group, Bar = 200 μm; **c** CCK-8 assay was used to detect the viability of VSMCs coincubated with exosomes harboring Over-circCEBPZOS or Sh-circCEBPZOS at 24 h, 48 h and 72 h; **d** Flow cytometry was used to detect the cell cycle of VSMCs coincubation with exosomes harboring with Over-circCEBPZOS or Sh-circCEBPZOS; **e** Transwell analysis was used to detect the migration of VSMCs coincubated with EVs harboring Over-circCEBPZOS or Sh-circCEBPZOS, Bar = 200 μm. Data are mean ± SD. **$p < 0.01$, Over-circCEBPZOS group vs NC group; ##$p < 0.01$, Sh-circCEBPZOS vs NC group.

also prefer to express in CMECs compared with cardiomyocytes and fibroblasts (Supplementary Fig. 1). Gain- or loss-of-function mutations in circCEBPZOS were achieved by coincubation with the EVs harboring with overexpression (Over) or suppression (Sh-) vectors of circCEBPZOS. qRT–PCR analysis showed that the expression of circCEBPZOS was successfully overexpressed or suppressed (Fig. 3a). Meaningfully, we noticed that the numbers of capillary-like structures were significantly decreased in Sh-circCEBPZOS-treated cultures and increased in the Over-circCEBPZOS-treated group, when compared to the NC group (Fig. 3b). Further analysis showed that the tubes formed by CMECs were significantly shorter in the Sh-circCEBPZOS group than in the NC group but elongated in the Over-circCEBPZOS group. These results indicated that circCEBPZOS could promote angiogenesis of CMECs. The results of an additional CCK-8 assay also demonstrated that the viability of VSMCs was significantly increased in the Over-circCEBPZOS group compared to the NC group, while it was significantly suppressed in cultures treated with Sh-circCEBPZOS at 48 h and 72 h (Fig. 3c). The results of cell cycle analysis showed that the number of VSMCs staged in the G1 phase was significantly decreased in the Over-circCEBPZOS-treated cultures, while the number of VSMCs progressed to the S and G2 phases was significantly increased in comparison with the NC group. Additionally, treatment with Sh-circCEBPZOS induced the opposite results compared to the NC group (Fig. 3d). In addition, we found that while Over-circCEBPZOS promoted VSMC migration, Sh-circCEBPZOS inhibited migration (Fig. 3e). Jointly, our novel results suggested that overexpression of circCEBPZOS promoted the angiogenesis of CMECs, and increased the proliferation and migration of VSMCs.

**circCEBPZOS acted as a molecular sponge for miR-1178-3p.** We used the CircInteractome program to predict with which miRNA the circCEBPZOS would interact. The results showed that

miR-614, miR-587, miR-1290 and miR-1178-3p had the highest Context+ Score and could bind with circCEBPZOS (Fig. 4a). Further analysis showed that the expression of miR-1178-3p was significantly elevated in EVs isolated from the serum of patients with postinfarct cardiac remodeling compared with those isolated from the non-postinfarct cardiac remodeling and control groups. However, the expression of miR-614 and miR-587 was significantly decreased in the EVs isolated from the serum of patients with postinfarct cardiac remodeling compared with the nonpostinfarct cardiac remodeling and control groups. The expression of miR-1290 was dramatically upregulated in the nonpostinfarct cardiac remodeling group but was not changed in the postinfarct cardiac remodeling group (Fig. 4b). To confirm this assumption, dual luciferase reporter analysis was performed. The results showed that the relative luciferase activity was significantly reduced in the circCEBPZOS-WT-pmiGLO group compared to the pmiGLO group treated with miR-1178-3p mimics. However, no significant changes were found in the circCEBPZOS-Mut-pmiGLO group, compared with the pmiGLO group (Fig. 4c). In addition, we noticed that the expression of miR-1178-3p was significantly decreased in Over-circCEBPZOS VSMCs but increased in Sh-circCEBPZOS VSMCs (Fig. 4d). These results demonstrated that circCEBPZOS is a direct target of miR-1178-3p.

**miR-1178-3p inhibited the angiogenesis of CMECs and the proliferation and migration of VSMCs.** Considering the demonstrated interaction between circCEBPZOS and the miR-1178-3p, we next confirmed the potential role of miR-1178-3p in CMECs angiogenesis and the proliferation and migration of VSMCs by assessing the loss and gain of miR-1178-3p functional mutations. We found that miR-1178-3p was also prefer to express in CMECs compared with cardiomyocytes and fibroblasts (Supplementary Fig. 1). The qRT–PCR analysis results showed that the expression of miR-1178-3p was successfully overexpressed or suppressed (Fig. 5a). Tube formation analysis showed that the

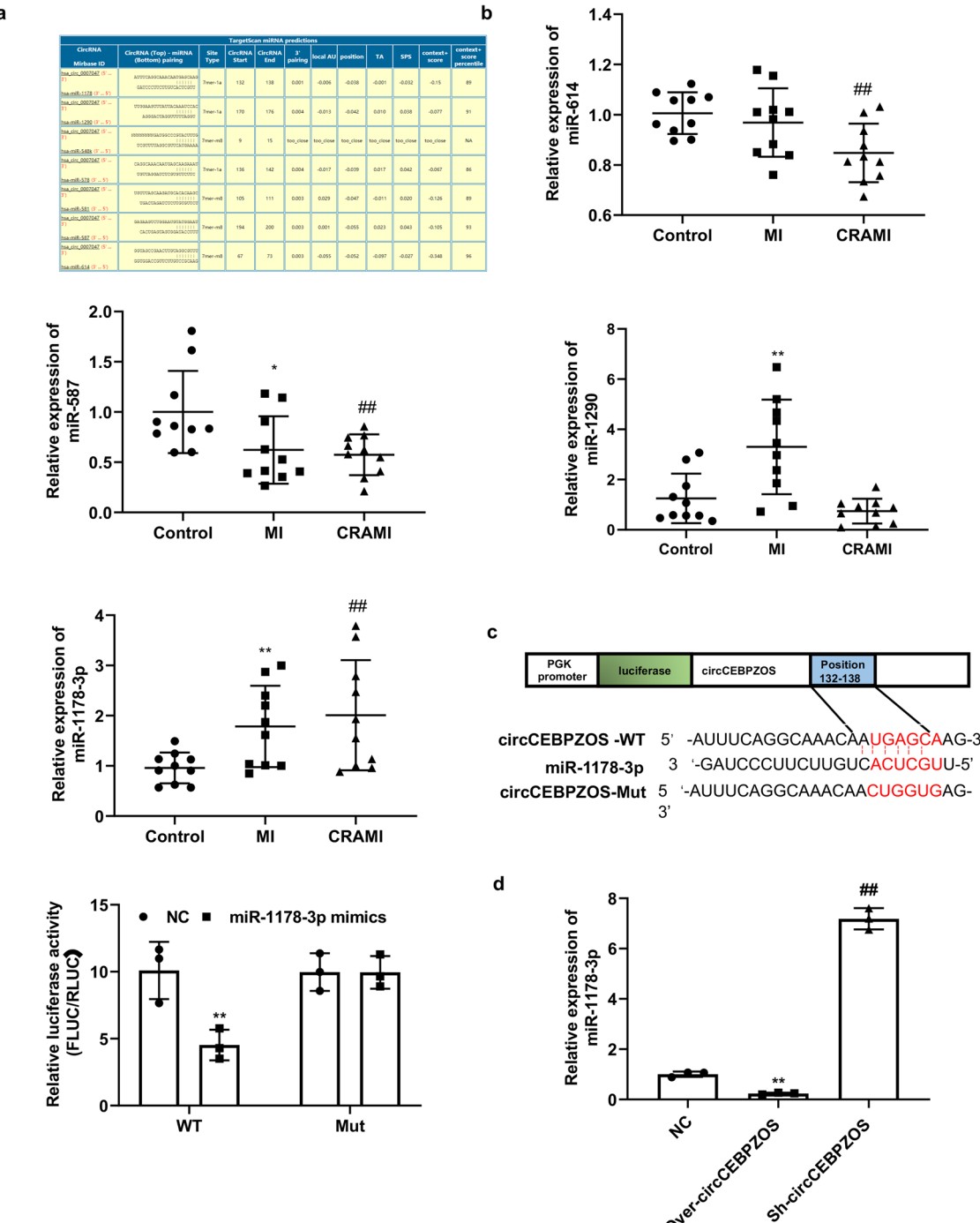

**Fig. 4 circCEBPZOS acts as a molecular sponge for miR-1178-3p. a** miRNA targets of circCEBPZOS were predicted by CircInteractome. **b** Expression of miR-614, miR-587, miR-1290 and miR-1178-3p in the EVs of serum from patients with and without postinfarct cardiac remodeling detected using qRT–PCR. Data are mean ± SD. **P < 0.01, N-CR group vs. Control group; ##P < 0.01, CR group vs. N-CR group. CR, postinfarct cardiac remodeling; N-CR, nonpostinfarct cardiac remodeling. **c** Dual luciferase reporter system analysis to detect the regulatory effect of miR-1178-3p on circCEBPZOS. **d** Expression of miR-1178-3p detected in VSMCs coincubated with EVs harboring with Over-circCEBPZOS or Sh-circCEBPZOS, U6 served as control. Data are mean ± SD. **P < 0.01, Over-circCEBPZOS group vs. NC group; ##P < 0.01, Sh-circCEBPZOS vs. NC group.

number of capillary-like structures was significantly decreased in the miR-1178-3p mimic group but increased in the miR-1178-3p inhibitor group compared to the NC group. Further quantification analysis showed that the tube length of CMECs was significantly shorter in the miR-1178-3p mimic group than in the NC group, while it was longer in CMECs treated with the miR-1178-3p inhibitor (Fig. 5b). The above results jointly indicated that miR-1178-3p could inhibit angiogenesis of CMECs.

The results of the CCK-8 assay demonstrated that the viability of VSMCs was significantly decreased in the miR-1178-3p mimic group compared to the NC group, while it was dramatically elevated in the miR-1178-3p inhibitor group (Fig. 5c). Cell cycle analysis showed that the number of VSMCs entering the G1 phase was significantly increased in the miR-1178-3p mimic-treated group, while the number of VSMCs entering the S and G2 phases was significantly decreased compared to the NC group.

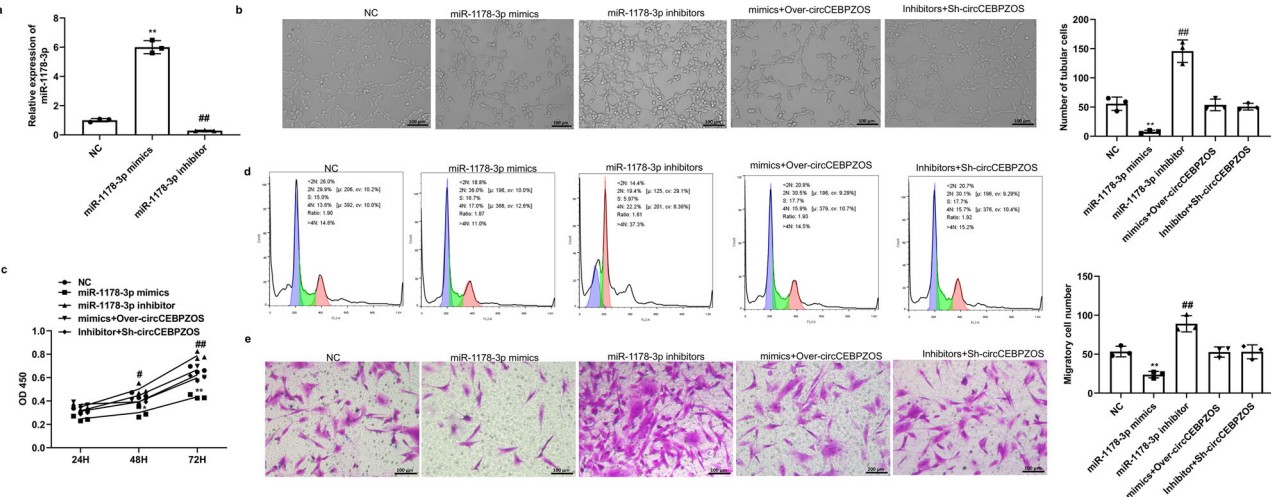

**Fig. 5 miR-1178-3p inhibited angiogenesis of CMECs and proliferation and migration of VSMCs. a** Expression of miR-1178-3p was detected using qRT–PCR in CMECs treated with miR-1178-3p mimics or inhibitors; **b** Tube formation analysis to detect capillary-like structures in the miR-1178-3p mimic group or miR-1178-3p inhibitor group and rescued by circCEBPZOS, Bar = 200 µm; **c** CCK-8 assay was used to detect the viability of VSMCs in the miR-1178-3p mimics or miR-1178-3p inhibitor group and rescued by circCEBPZOS at 24 h, 48 h and 72 h; **d** Flow cytometry was used to detect the cell cycle of VSMCs in the miR-1178-3p mimics or miR-1178-3p inhibitor group and rescued by circCEBPZOS; **e** Migration of VSMCs was detected by Transwell assay when miR-1178-3p was overexpressed or suppressed and rescued by circCEBPZOS, Bar = 200 µm. Data are mean ± SD. **$P < 0.01$, miR-1178-3p mimics group vs. NC group; ##$P < 0.01$, miR-1178-3p inhibitor group vs. Control group.

Meaningfully, the opposite results were detected in cultures of VSMCs treated with miR-1178-3p inhibitor (Fig. 5d). Furthermore, we showed that overexpressing the miR-1178-3p in cultured VSMCs inhibited their migration. In contrast, suppressing miR-1178-3p expression promoted the migration of cultured VSMCs (Fig. 5e). In addition, we found that the inhibitory effects of miR-1178-3p on the angiogenesis of CMECs and the proliferation and migration of VSMCs could be partially reversed by the coincubation of these cultured cells with EVs harboring the overexpressed circCEBPZOS or Sh-circCEBPZOS plasmid in the miR-1178-3p mimic or inhibitor groups, respectively (Fig. 5b–e). In order to further confirmed that circCEBPZOS affected the function of miR-1178-3p on the angiogenesis of CMECs and the proliferation and migration of VSMCs, Rescue experiment by overexpression of circCEBPZOS with mutant miR-1178-3p binding site was performed. The results showed that the function of miR-1178-3p mimics on the angiogenesis of CMECs and the proliferation and migration of VSMCs could not been reversed when overexpression of circCEBPZOS with mutant miR-1178-3p binding site (Supplementary Fig. 2). These findings suggested that overexpression of miR-1178-3p inhibited CMECs angiogenesis, and VSMC proliferation and migration.

**PDPK1 is identified as the target of miR-1178-3p.** To identify the downstream target of miR-1178-3p, three online databases, TargetScan7.2, miRwalk and miRDB, were used to predict its targets[19–21]. As shown in Fig. 6a, we found three common targets from the three databases, including cholinergic receptor nicotinic beta 4 (CHRNB4), Stonin 2 (STON2) and PDPK1. Subsequent qPCR and western blotting demonstrated that the expression of PDPK1 was significantly downregulated while the expression of CHRNB4 and STON2 was upregulated in the serum of the CR group compared with the N-CR and control groups (Fig. 6b, c). The expression of PDPK1 was most downregulated among the three genes and was selected for further study. Furthermore, the consecutive results of the luciferase reporter assay indicated that cotransfection of the PDPK1-WT pmiGLO reporter plasmids and miR-1178-3p mimic predominantly reduced the luciferase activity. Conversely, cotransfection of the PDPK1-Mut pmiGLO

reporter plasmids with miR-1178-3p mimic showed no obvious effect on luciferase activity (Fig. 6d). In addition, we found that miR-1178-3p overexpression reduced PDPK1 expression while miR-1178-3p suppression increased PDPK1 expression at both the mRNA and protein levels (Figs. 6e, f). These results indicated that PDPK1 is targeted by miR-1178-3p.

**miR-1178-3p-mediated PDPK1 regulates CMECs angiogenesis and VSMCs proliferation and migration.** The next set of experiments was aimed at exploring how the loss or gain of PDPK1 expression would affect the functions of the cultured cells. The results of qRT–PCR and western blotting analysis showed that PDPK1 could be successfully overexpressed or suppressed in cultured CMECs (Fig. 7a, b). Consecutive tube formation analysis indicated that PDPK1 overexpression significantly increased the number of capillary-like structures and tube length, while PDPK1 suppression greatly decreased the number of capillary-like structures and CMEC tube length compared to those of the control group (Fig. 7c).

The CCK-8 assay demonstrated that the viability of VSMCs was significantly elevated in the Over-PDPK1 group compared to the NC group, while it was dramatically decreased in the Si-PDPK1 group after 48 h and 72 h incubations (Fig. 7d). Cell cycle analysis further indicated that the number of VSMCs in G1 phase was significantly decreased in the Over-PDPK1 group, while the number of VSMCs that entered the S and G2 periods was significantly increased compared to the NC group. Meaningfully, the opposite results were detected in the Si-PDPK1 group compared to the NC group (Fig. 7e). Furthermore, we showed that the overexpression of PDPK1 could promote the migration of VSMCs, while the suppression of PDPK1 inhibited the migration of VSMCs (Fig. 7f). In addition, we found that the effects of PDPK1 on the angiogenesis of CMECs and the proliferation and migration of VSMCs could be reversed by adding miR-1178-3p mimics to the PDPK1 overexpression group or by treating the si-PDPK1 group with the miR-1178-3p inhibitors (Fig. 7c–f). In addition, we found that the function of overexpressing circCEBPZOS on CMECs angiogenesis and VSMCs proliferation and migration could be reversed by

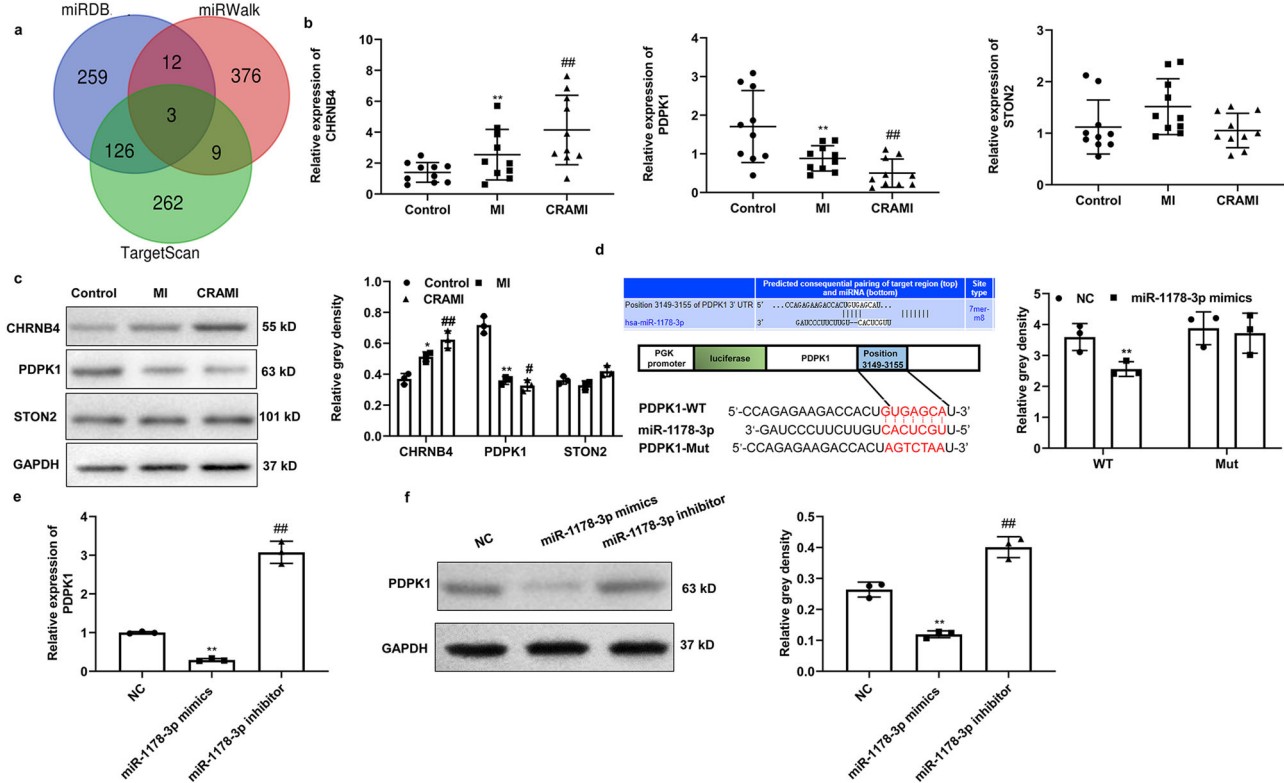

**Fig. 6 PDPK1 is the target of miR-1178-3p. a** Targets of miR-1178-3p were predicted by the online databases TargetScan7.2, miRwalk and miRDB; **b**, **c** The qPCR and western blotting analysis were used to detect the expression of PDPK1, CHRNB4 and STON2 in the serum of patients with and without postinfarct cardiac remodeling, respectively. GAPDH served as control. Data are mean ± SD. **P < 0.01, N-CR group vs. Control group; ##P < 0.01, CR group vs. N-CR group. CR, postinfarct cardiac remodeling; N-CR, nonpostinfarct cardiac remodeling. **d** Luciferase reporter assays to verify the binding of miR-1178-3p to PDPK1 at the 3'UTR; **e**, **f** Expression of PDPK1 was detected using qRT–PCR and western blotting in VSMCs treated with miR-1178-3p mimics or inhibitors at the mRNA and protein levels, respectively. Data are mean ± SD. **P < 0.01, miR-1178-3p mimics group vs. NC group; ##P < 0.01, miR-1178-3p inhibitor group vs. NC group.

administration of sh-PDPK1 (Supplementary Fig. 3). Altogether, these results indicated that circCEBPZOS-miR-1178-3p axis induced PDPK1 regulated the angiogenesis of CMECs and the proliferation and migration of VSMCs.

**circCEBPZOS alleviated postinfarct cardiac remodeling in vivo.** To further confirm the beneficial effects of the circCEBPZOS in postinfarct ventricular remodeling, EVs containing circCEBPZOS overexpressing or suppression plasmid were injected into the tail vein of mice after induction of their experimental MI. We first examined the toxicity of EVs to the liver, lung and muscle. H&E staining assays showed that the EVs had no by effect to liver, lung or muscle (Supplementary Fig. 4), suggesting that EVs are safe for injection. Furthermore, we found that the expression of circCEBPZOS was significantly downregulated compared with that in the sham group. However, the expression of circCEBPZOS was successfully overexpressed or suppressed when treated with EVs overexpressing or suppressing the circCEBPZOS plasmid in the MI mice (Fig. 8a). Importantly, we recorded that the injection of EVs containing Over-circCEBPZOS significantly decreased LVEDD and LVESD and improved LVEF and LVFS (P < 0.05). In contrast, the injection of EVs containing Sh-circCEBPZOS into parallel MI mice induced the opposite effects (Fig. 8b, Supplementary Table 1). Even the initial comparison of the myocardial sections from all experimental groups, stained with H&E, clearly illustrated the exclusive beneficial effects of treatment with the OvercircCEBPZOS. Analysis of the parallel histologic sections stained with the Masson's

method or with the Sirius Red further confirmed a highly reduced degree of cardiac fibrosis in the myocardium of the Over-circCEBPZOS Exo-treated group compared to the untreated MI group. In contrast, we noticed that administration of EVs of Sh-circCEBPZOS aggravated postinfarct ventricular remodeling (Fig. 8c).

Further histologic analysis also demonstrated that treatment with the Over-circCEBPZOS EVs led to a significant increase in capillary density compared to the untreated MI group. In contrast, the density of capillaries was significantly decreased in the Sh-circCEBPZOS EV group (Fig. 8d). The western blotting analysis and immunofluorescence staining assay results for the parallel tissue samples additionally demonstrated that the expression of HIF-1α, VEGF and VEGFR was significantly elevated in the Over-circCEBPZOS EVs-treated group, while decreased in the Sh-circCEBPZOS EV-treated group compared with the NC group (Fig. 8d, e). In fact, we found that the similar effects could be achieved by transfection with the vectors without EVs (Supplementary Fig. 5, Supplementary Fig. 6). These results were also confirmed by immunofluorescence staining assays. These results demonstrated that circCEBPZOS functions as an effective suppressor of the post-MI deleterious ventricular remodeling.

The next series of experiments aimed to investigate whether circCEBPZOS could affect the expression of miR-1178-3p and PDPK1 in heart tissues. Meaningfully, we found that administration of EVs containing an overexpressing circCEBPZOS plasmid greatly decreased the levels of miR-1178-3p and increased PDPK1 levels. However, treatment with EVs

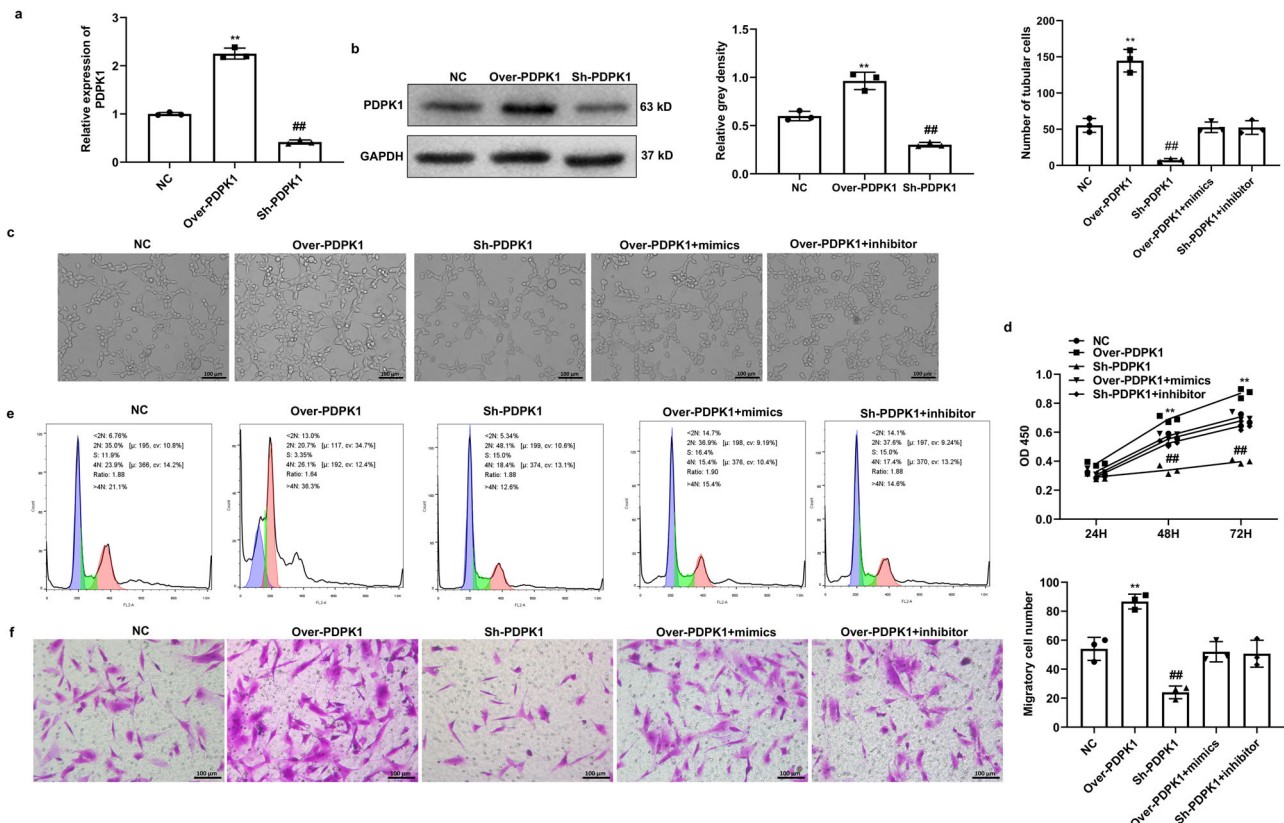

**Fig. 7 miR-1178-3p-mediated PDPK1 regulated angiogenesis of CMECs and proliferation and migration of VSMCs. a, b** Expression of PDPK1 was detected by qRT–PCR and western blotting by overexpressing or suppressing of PDPK1 in CMECs, respectively, GAPDH served as a control; **cc** Tube formation analysis was used to detect capillary-like structures in the PDPK1 overexpression or suppression group and rescued by miR-1178-3p; Bar = 200 µm; **d** CCK-8 assay was used to detect the viability of VSMCs in the PDPK1 overexpression or suppression of group and rescued by miR-1178-3p at 24 h, 48 h and 72 h; **E** Flow cytometry was used to detect the cell cycle of VSMCs in the PDPK1 overexpression or suppression group and rescued by miR-1178-3p; **f** Migration of VSMCs was detected by Transwell assay when PDPK1 overexpressed or suppressed and rescued by miR-1178-3p, Bar = 200 µm. Data are mean ± SD. **$P < 0.01$, Over-PDPK1 group vs. NC group; ##$P < 0.01$, Sh-PDPK1 group vs. NC group.

containing circCEBPZOS siRNA significantly increased the expression of miR-1178-3pm and decreased PDPK1 levels in vivo (Fig. 9a, b). Additional western blotting and immuno-fluorescence staining also confirmed the previously mentioned expression changes of PDPK1 in different groups (Fig. 9c, d). In addition, ELISA also found that the content of PDPK1 was significantly decreased in the MI group. In contrast, the content of PDPK1 was significantly increased but further decreased when EVs overexpressing or suppressing circCEBPZOS were treated in the MI group (Fig. 9e). Therefore, these results demonstrated that circCEBPZOS attenuated post-MI ventricular remodeling via regulation of the miR-1178-3p/PDPK1 axis in vivo.

## Discussion

In the present study, EV-derived circCEBPZOS was identified as a novel regulator of postinfarct ventricular remodeling. Functional and mechanistic analysis revealed that EV-derived cir-cCEBPZOS alleviated adverse postinfarct cardiac remodeling by promoting angiogenesis by regulating the miR-1178-3p/PDPK1 axis.

Cardiac remodeling after MI refers to a cascade of structural and functional changes in cardiomyocytes and intercellular substances. An extremely complex pathogenesis of this process includes myocardial hypertrophy, fibrosis, inflammation, autophagy and metabolic malfunction[22]. It has been recently demonstrated that EVs are considered the main mediators of intercellular communication in the myocardium, which may

attenuate the development of detrimental structural changes and the consequent heart failure after MI[23]. Thus, the presence of EVs likely contributes to the inhibition of fibrosis, promotion of angiogenesis and alleviation of inflammation and pyroptosis by translocation of intercellular molecules, proteins, and organelles. Recently, particular attention has been focused on EV-derived circRNAs since the discovery that the circular form of noncoding RNAs, such as circFndc3b, circRNA HIPK3 and circRNA CDR1, could contribute to the beneficial regulation of post-MI ventricular remodeling[24,25]. Further studies exploring the mechanistic involvement of diverse EVs in the myocardial healing demonstrated that hypoxia-elicited mesenchymal stem cell-derived EVs facilitated cardiac repair through miR-125b-mediated amelioration of apoptosis and pyroptosis of cardiomyocytes within the infarct region of a murine MI model[26].

The results of other studies also indicated that angiogenesis plays a crucial role in ameliorating adverse myocardial remodeling, thereby contributing to the improvement of cardiac function and preventing the heart failure[27]. Silencing of the angiogenesis inhibiting factor epiregulin disrupted ERK1/2 signaling and promoted LV remodeling[28]. Inhibition of miR-17 prevented high glucose-induced impairment of angiogenesis in diabetic mice, thereby improving their cardiac function after MI by targeting VEGFA[29]. Moreover, vildagliptin, granulocyte colony stimulating factor (G-CSF) and oncostatin M (OSM) have been listed as angiogenic factors contributing to the improvement of cardiac function after MI[30,31].

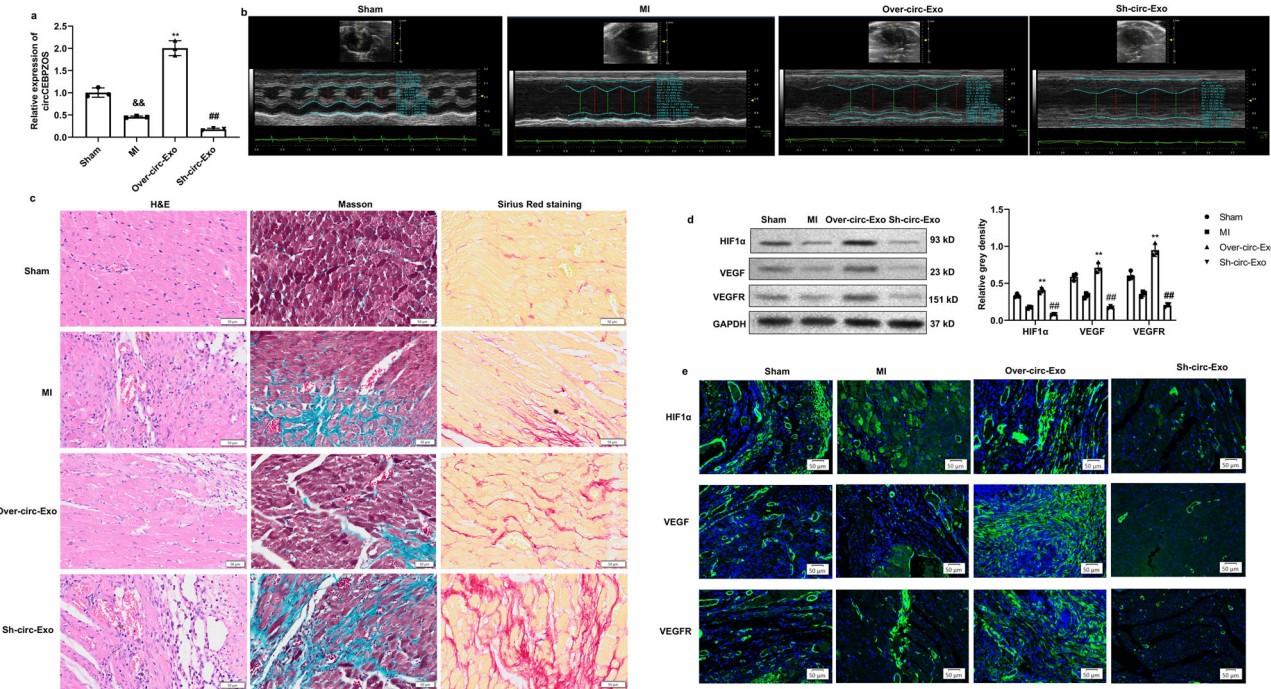

**Fig. 8 circCEBPZOS alleviated postinfarct cardiac remodeling. a** qRT–PCR was used to verify the expression of circCEBPZOS in heart tissue treated with exosomal circCEBPZOS. GAPDH served as a control. **b** Echocardiographic measurements to detect the in the model group compared with sham group; **c** Pathological changes detected by HE, Masson and Sirius Red staining in heart tissue treated with circCEBPZOS, Bar = 50 μm; **d** Western blotting analysis to detect the expression of HIF-1α, VEGF and VEGFR in the Over-circa-Exo group or Sh-circa-Exo group. GAPDH served act as control; **e** Immunofluorescence staining was used to detect the expression of HIF-1α, VEGF and VEGFR in the Over-circa-Exo group or Sh-circa-Exo group. DAPI was used to stain the nuclei; Bar = 50 μm. Data are mean ± SD. **p < 0.01, Over-circ-Exo group vs. MI-NC group; ##p < 0.01, She-circ-Exo group vs. MI-NC group.

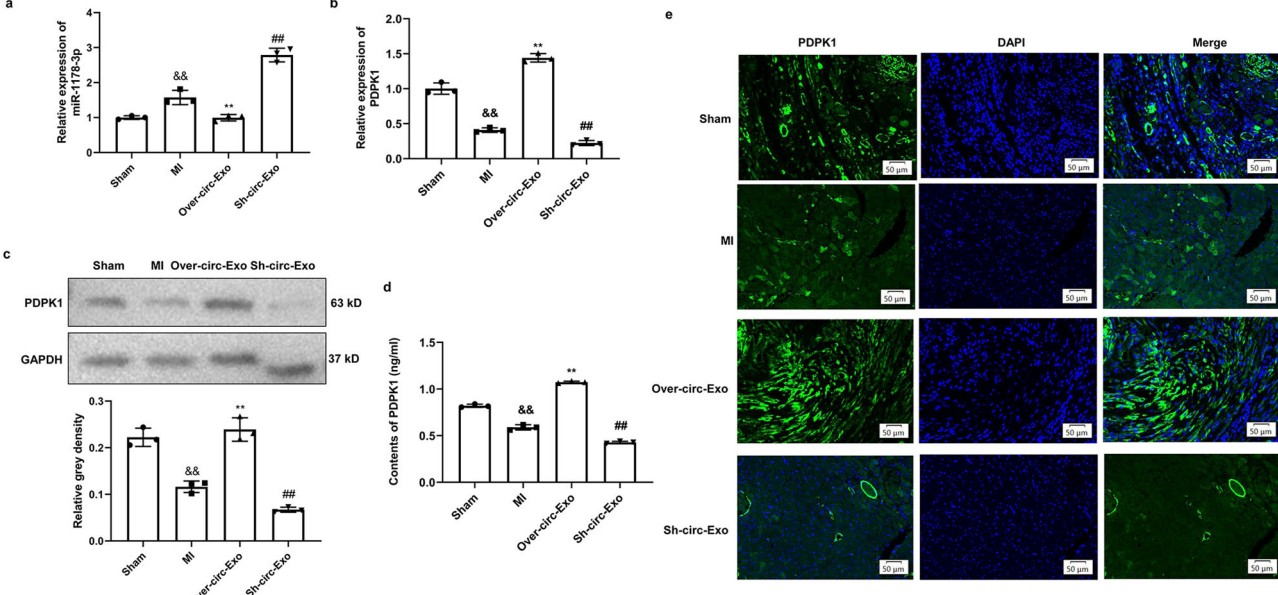

**Fig. 9 Expression changes of miR-1178-3p and PDPK1 in vivo treatment with exosomes harboring circCEBPZOS. a** Expression of miR-1178-3p was detected by qRT–PCR, U6 acted as a control; **b**, **c** Expression of PDPK1 was detected by qRT–PCR and western blotting. GAPDH served as control. **d** Immunofluorescence staining was used to detect the expression of PDPK1 in heart tissues. DAPI was used to stain the nuclei; Bar = 50 μm. **e** ELISA was used to detect the contents of PDPK1.Data are mean ± SD. **P < 0.01, Over-circ-Exo group vs. Model-NC group; ##P < 0.01, She-circ-Exo group vs. Model-NC group.

On the other hand, numerous studies have demonstrated that EVs play a regulatory role in angiogenesis during cardiac remodeling after MI by harboring certain noncoding RNAs or metabolic proteins[32]. Thus, EVs derived from Akt-modified human umbilical cord mesenchymal stem cells improve cardiac regeneration and promote angiogenesis by activating platelet-derived growth factor D[33]. Exosomal miR-132 and miR-146a delivered by mesenchymal stem or adipose-derived stem cells attenuated ischemic myocardial damage by inducing angiogenesis in MI[34,35]. Importantly, the results of our present study revealed that the newly identified EV-derived circCEBPZOS, which promoted angiogenesis, proliferation, and migration of VSMCs in vitro and could alleviate ventricular remodeling in a murine MI model.

HIF-1α, VEGF and VEGFR have been previously indicated as angiogenetic factors after MI[36,37]. We found that the expression of these factors in the murine myocardium subjected to experimental MI was significantly elevated after treatment with Over-circCEBPZOS EVs but decreased after treatment with Sh-circCEBPZOS EVs. Therefore, these results demonstrated that circCEBPZOS functions as a potential suppressor of the adverse postinfarct cardiac remodeling by promoting angiogenesis.

In the present study, bioinformatics analysis and a dual luciferase reporter system demonstrated that miR-1178-3p not only binds to circCEBPZOS but also to PDPK1, the leucyl-specific aminopeptidase activated factor, contributing to the VEGF-dependent activation of S6K and the ultimate stimulation of endothelial cell proliferation and angiogenesis[38]. Interestingly, we also established that the overexpression of miR-1178-3p inhibited angiogenesis, proliferation, and migration of VSMCs, suggesting that miR-1178-3p could inhibit overzealous angiogenesis in cardiac remodeling after MI. On the other hand, it has been shown that the ablation of PDPK1 in cultured vascular endothelial cells enhanced insulin sensitivity and suppressed angiogenesis[39]. These results demonstrated that PDPK1 acts as a positive regulator in angiogenesis. Consistent with these findings, we revealed that overexpression of PDPK1 promoted angiogenesis, proliferation and migration of VSMCs. In addition, we found that the function of miR-1178-3p and PDPK1 on the angiogenesis, proliferation and migration of VSMCs could be reversed by circCEBPZOS and miR-1178-3p, respectively. Altogether, these results indicated that circCEBPZOS alleviated postinfarct ventricular remodeling by promoting angiogenesis via miR-1178-3p-mediated PDPK1 expression.

In this present study, we identified a novel EV-derived circCEBPZOS that was profoundly downregulated in patients with detrimental ventricular remodeling after MI. Meaningfully, we further established that EV-derived circCEBPZOS alleviated postinfarct remodeling by promoting angiogenesis and expanding the collateral network by regulating miR-1178-3p/PDPK1 axis, which consequently preserved the function of the border zone of the infarcted myocardium. These findings suggest that over-expressing or application of EV-derived circCEBPZOS should be further investigated as a putative therapeutic target in ischemic heart failure patients.

## Methods

**Clinical specimens and ethical statement**. Clinical serum specimens were obtained from MI patients with ($n = 10$) or without ($n = 10$) cardiac remodeling at Guangdong Provincial Hospital of Chinese Medicine following the inclusion and exclusion criteria (Supplementary Table 2). The group of healthy individuals ($n = 10$) matched with our patients, served as an additional control. All participants of our study signed informed consent forms, and the entire project was approved by the ethics committee of Guangdong Provincial Hospital of Chinese Medicine (B2015-129-01) and conducted in accordance with the Declaration of Helsinki and its text revisions. The clinical parameters of each group are summarized in Supplementary Table 3.

**Isolation and identification of human EVs**. EVs were isolated from healthy human serum by using ExoQuick EV precipitation solution (SBI, CA, USA) following the user manual. Briefly, serum was centrifuged at 3000 g for 15 min, 63 μL ExoQuick EVs precipitation solution was added, and the mixture was refrigerated for 30 min at 4 °C. Then, the EVs were resuspended in 100 μL of sterile PBS after centrifugation at 1500 g for 30 min. Particle size, morphology and the total amount of the EVs were then identified by TEM (Philips TECNAI 20, Netherland) and by the NTA, respectively. The EV protein markers CD6 and CD9 and tumor susceptibility gene 101 (TSG101) were identified by western blot assay[40].

**Cell cultures and transfection of human VSMCs and CMECs**. Human VSMCs and cardiac microvascular endothelial cells (CMECs) were purchased from the American Type Culture Collection (ATCC, Bethesda, MD, USA). VSMCs were cultured in 90% Dulbecco's modified Eagle's medium (DMEM, 12430054, Gibco, USA) supplemented with 10% fetal bovine serum (FBS, Gibco, USA), 100 ug/mL penicillin and 100 ug/mL streptomycin in a 5% $CO_2$-containing incubator under 95% saturation humidity at 37 °C. For the functional analysis of circCEBPZOS, the overexpression or suppression plasmids of circCEBPZOS were coincubated with serum EVs and then coincubated with VSMCs. For further analysis of the effects of phosphoinositide-dependent kinase-1 (PDPK1) or miR-1178-3p, 60–80% confluent cell cultures were transfected with pCDNA3.1-PDPK1 overexpression or suppression plasmids and with miR-1178-3p mimics or inhibitor using Lipofectamine^TM 2000 transfection reagents (52887; Invitrogen, USA). Quantitative reverse transcription-polymerase chain reaction (qRT–PCR) and western blots were additionally used to detect the overexpression or silencing efficiency.

**Quantitative reverse transcription-polymerase chain reaction**. The expression of circCEBPZOS, PDPK1 and miR-1178-3p in cells or tissues were detected by qRT–PCR. Total RNA from tissues and cells was extracted by TRIzol reagent (15596026, Invitrogen, Carlsbad, CA, USA). For the identification of circCEBPZOS, 2 μg samples of total RNA were incubated for 30 min at 37 °C with or without 2.5 U of RNase R (Epicentre Technologies, Madison, WI). Then, reverse transcription reactions were performed using cDNA using HiScript III RT SuperMix for qPCR (+gDNA wiper) (R323-01, Vazyme, Nanjing, China) (2 μg RNA samples) according to the manufacturer's instructions. qRT–PCR were also performed using an ABI 7500 instrument (ABI7500, ABI, USA) with SYBR Green Mix (4913914001, Roche) in a 20-μL reaction system containing 9 μL SYBR Mix, 0.5 μL of each primer (10 μM), 2 μL cDNA template and 8 μL RNase free $H_2O$. The following programs were used: 95 °C for 10 min, followed by 40 cycles at 95 °C for 15 s, 60 °C for 1 min, and 72 °C for 45 s. Relative expression levels of targeted genes were calculated using the $2^{-\triangle\triangle Ct}$ method, which was normalized to GAPDH (for PDPK1 and circCEBPZOS) and to U6 (for miR-1178-3p). Nuclear and cytoplasmic RNA was extracted using a nuclear and cytoplasmic RNA purification kit (Fisher scientific, Hampton, USA). The samples of DNA or cDNA were amplified by divergent or convergent primers of circCEBPZOS, respectively. The amplification process were run as follows: 95 °C for 5 min, followed by 32 cycles at 94 °C for 15 s, 60 °C for 30 min, and 72 °C for 30 s. The final products were then detected by 1% Agarose gel electrophoresis. All primers used in the present study were synthesized by General Biol company (General Biol Co., Ltd, Anhui, China), and the detailed information is listed in Supplementary Table 4.

**Western blotting**. Total protein was extracted from the serum or heart tissues using RIPA lysis buffer (P0013, Beyotime, Shanghai, China) according to the manufacturer's instructions[41]. Twenty grams of protein aliquots derived from each experimental samples, were assessed with the BCA protein assay kit (70-PQ0012, MultiSciences, China). They were boiled at 100 °C for 5 min, separated using 10–12% SDS-PAGE electrophoresis and transferred onto PVDF membranes. The membranes were then blocked with 5% lipid-free milk/TBST buffer for 2 h at room temperature, and then incubated with anti-PDPK1 antibody (ab234064, 1:1000, Abcam, UK), with anti-HIF1a (20960-1-AP, 1:200, Proteintech, USA), with anti-VEGF (19003-1-AP, 1:1000, Proteintech, USA), with anti-VEGFR (26415-1-AP, 1:1000, Proteintech, USA), with anti-CD6 (ab109217, 1:1000, Abcam, UK), with anti-CD9 (ab92726, 1:500, Abcam, UK), with anti-TSG101(ab125011, 1:1000, Abcam, UK) and with anti-GAPDH (ab32391, 1:1000, Abcam, Cambridge, UK) at 37 °C for 2 h, respectively. Then, all blotting membranes were subsequently incubated with secondary anti-mouse IgG antibodies (ab205719, 1:20000, Abcam, Cambridge, UK) for 1–2 h. All the immuno-complexes were finally detected using ECL after washing with TBST and analyzed using the Image-Pro Plus 6.0 software.

**Cell proliferation and cell cycle assays**. The cell proliferation ability of VSMCs was evaluated by the CCK-8 Counting Kit (A311-02, Vazyme, Nanjing, China) according to the manufacturer's instructions. Briefly, VSMCs transfected for 48 h were isolated by the enzymatic digestion and seeded in 96-well plates for 24, 48 and 72 h, respectively. Cultures were then incubated with medium containing 10 μL CCK-8 marker solution for 2 h at 37 °C. Absorbance was detected at 450 nm using a microplate reader. Cell cycle progressions were detected in cultures, fixed overnight in 70% ethanol at 4 °C by Flow cytometry (FACSCalibur, Becton) at 488 nm, after staining with propidium iodide (PI, KeyGEN Biotech, Nanjing, China).

Obtained results were then analyzed with ModFit LT software (Verity software House).

**Cell migration assays**. VSMCs from different groups were harvested after culturing in serum-free medium for 16 h, resuspended in serum-free medium and transferred to two-layered Transwell chambers pre-coated with Matrigel (BD Biosciences) and incubated at 37 °C for the following 24 h. Both membranes were then fixed with 100% methanol and stained with 1% toluidine blue. The numbers of cells attached to each membrane were counted under a light microscope (Zeiss710, Germany).

**Tube formation assay**. The growth factor reduced Matrigel (BD Biosciences) was thawed on ice and the 300 µL samples of this preparation were plated into 24-well plates and incubated for 30 min at 37 °C to allow polymerization. CMECs were suspended in 0.2% endothelial growth basal medium (EBM), and $5 \times 10^4$ cells of CMECs were added to Matrigel-coated wells. To assess the potential influence of circCEBPZOS, miR-1178-3p and PDPK1 in VSMCs, cultures of CMECs were co-incubated with EVs contained overexpressing or silencing circCEBPZOS, with miR-1178-3p mimics/inhibitors or overexpressing/silencing the PDPK1 plasmids. Then, the influence of these diverse preparations on the initiation of cellular tubes formation was monitored for 12 h at 37 °C under a phase contrast microscope (×4) (Nikon TS100). Tube lengths were quantified using the Image J software (National Institutes of Health).

**Luciferase reporter assay**. For circCEBPZOS, the miRNA targets of circCEBPZOS was predicted by CircInteractome. For the downstream regulator of miR-1178-3p, online databases TargetScan 7.2, miRwalk and miRDB were used to predict the targets of miR-1178-3p. And then the fragment of circCEBPZOS or PDPK1 were cloned into the pmiGLO vector containing the wild-type sequence and mutant binding sequence, respectively. The circCEBPZOS or PDPK1 pmiGLO vector and miR-1178-3p mimcis were co-transfected into the VSMCs when the confluence reached at 60-70% by using the Lipofectamine™ 2000 transfection reagents (52887, Invitrogen, USA). Cells were washed twice with phosphate buffered saline and lysed using the passive lysis buffer after cultured for 48 h after transfection. The luciferase activity was evaluated using the Dual-Luciferase Reporter Assay System (Progema). The primers used for vector construction were listed in Supplementary Table 4.

**Model of experimental MI in mice**. Wild-type male C57BL/6 J mice (10–12 weeks weighing 24–27 g) were purchased from the Experimental Animal Center of Guangdong Province (Guangzhou, China) and maintained in the SPF Animal Center of Guangdong Provincial Hospital of Chinese Medicine under 12-h light-dark cycles at 21 ± 2 °C with a humidity of 50–80%. They were randomly divided into three groups with 15 mice per group. Then left anterior descending branches of their coronary artery were ligated to induce the acute experimental MI model as previously described[42]. The coronary arteries of mice from the sham group were only surgically stranded without ligation. Echocardiographic and histopathological changes were analysed to assess the cardiac remodeling 28 days after surgical procedures. EVs containing the overexpressing or suppressed circCEBPZOS plasmids were isolated and injected via a tail vein of mice from all different groups, 24 h after the surgical induction of theirs MI. Then, levels of their circCEBPZOS transfection's efficiency were assessed by qRT-PCR. The project was approved by the ethics committee of the Guangdong Provincial Hospital of Chinese Medicine.

**Echocardiography**. Echocardiographic assessment was performed as previously described assess the relevant heart's actions[43]. Briefly, the LV function in C57BL/6 J mice was conducted just before sacrifice using the Acuson Sequoia C512 system equipped with a 15L8 linear array transducer with 30 MHz. Mice were anesthetized with 1.5% isoflurane mixed with oxygen and placed in a supine position on a heating pad. Short-axis measurements were used to capture M-mode tracing at the level of the papillary muscles with a 25-mm signal depth. Three to six consecutive cardiac cycles were measured using M-mode tracings with the accompanying software.

**Histopathological changes analysis**. Hematoxylin and eosin (H&E), Masson and Sirius red staining were performed to evaluate the histopathological changes in vivo[44]. Briefly, heart tissue was sliced into 8-µm-thickness sections and fixed with 4% paraformaldehyde at room temperature. The tissue sections were then stained with H&E Staining Kit (C0105S, Beyotime, China), Masson staining solution (R20379, YuanYe Biotech, Shanghai, China) and Sirius-Red staining (2610-10-8, Maokang Biotech, Shanghai, China), respectively. The sections were dehydrated and sealed with neutral gum and subsequently washed with running water and imaged under a microscope (Olympus, Japan). Three infarct areas were taken from each mouse from >three independent mice.

**Immunofluorescence staining**. Immunofluorescence staining of the parallel histologic slides was used to detect the expression of proteins involved in angiogenesis, including hypoxia-inducible factor 1 (HIF-1α), vascular endothelial growth factor

(VEGF) and vascular endothelial growth factor receptor (VEGFR). Briefly, 4-µm serial microtome sections of the paraffin embedded heart tissues were generated and placed in 3% catalase for 15 min and blocked with 50 µL goat serum for 20 min at room temperature. Subsequently, the serial sections were incubated with anti-PDPK1 antibody (ab234064, 1:1000, Abcam, UK), with anti-HIF1a (20960-1-AP, 1:200, Proteintech, USA), with anti-VEGF (19003-1-AP, 1:1000, Proteintech, USA), with anti-VEGFR (26415-1-AP, 1:1000, Proteintech, USA) at 37 °C for 2 h, respectively. The parallel sections were consecutively incubated with the FITC conjugated goat anti-rabbit IgG (ab6785, Abcam, Cambridge, UK, 1:1000) at 37 °C for 1 h, then with 50 µl of DAPI dye in the dark at room temperature for 5 min for the nuclei counterstain. The sections were accessed using an XSP-36 microscope (Boshida Optical Co., Ltd., Shenzhen, China). Three infarct areas were taken from each mouse from >three independent mice.

**Statistical analysis**. Data are presented as the mean ± standard deviation (SD) from at least three independent experiments performed in triplicate. Analyses were performed using Prism 8.1.2 (GraphPad Software Inc.) by unpaired $t$-test and one-way analysis of variance (ANOVA) between two groups and >two groups, respectively. $P$-values < 0.05 were defined as the level of statistical significance.

**Reporting summary**. Further information on research design is available in the Nature Portfolio Reporting Summary linked to this article.

## Data availability

All data generated or analysed during this study are included in this published article (and its supplementary information files). Uncropped and unedited blots (Figs. 1, 6, 7, 8, 9) were included in the Supplementary Fig. 7.

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

## Acknowledgements

We would like to thank all of the members of the State Key Laboratory of Dampness Syndrome for their critical and thoughtful suggestions. This study was funded by National Science Foundation (grant No. 82104962 & No. 82274271), Science and Technology Planning Project of Guangzhou (202102010301), Featured Innovative Project from Guangdong Provincial Universities (2019KTSCX029), Young Talents Support Project from China Association of Chinese Medicine (2019-QNRC2-C06), Team of prevention and treatment of acute myocardial infarction with Chinese medicine (2019KCXTD009), Foundation of Guangdong Province of CM (20211187&20201142). The sponsors have had no role in the project development, in the collection of data, in the preparation of this manuscript, nor the decision to publish.

## Author contributions

S.M. and A.H. drafted this manuscript; S.M., Y.L. and M.Z. performed the experiments, L.Y. made statistical analysis; P.C.Y. made a critical revision of the manuscript and contributed to the rationalization of the study. All authors read and approved the final manuscript.

## Competing interests

The authors declare no competing interests.
