## [Peer Review File · Communications Biology]

Reviewers' comments:

Reviewer #1 (Remarks to the Author):

I would recommend a substantial revision addressing my following concerns

The current manuscript by Shuai et al., titled "Exosomal hsa_circ_0007047 attenuates post-myocardial infarction remodeling by promoting angiogenesis via miR-1178-3p/PDPK1 axis" extensive experimentation and suggest that hsa_circ_0007047 promoted angiogenesis both in vitro and in vivo. Importantly overexpression of hsa_circ_0007047 in mice significantly improved fibrosis and cardiac function post-MI in mouse models. However, I have the following concerns.

Major concerns

How efficient was the plasmid able to include/reduce hsa-circ_0007047 expression in the mouse? And for how long

Authors should demonstrate which cells in the heart takes up these plasmids?

Does a human ortholog of hsa_circ_0007047 exist in mice? If yes, what happens to its levels post-MI mice?

Does over-expression plasmids for hsa-circ_0007047 affect miR-1178 levels in the heart post-MI?

Also, in general, what are miR-1178 levels post-MI in mouse hearts without hsa-circ_0007047 treatment.

hsa_circ_0007047 has how many binding sites for miR-1178-3p; authors should mention this at some point.

Circinteractome shows hsa_circ_0007047 has binding sites for the following miRs, what was the rationale in choosing miR-1178?

hsa-miR-1178 1

hsa-miR-1290 1

hsa-miR-548k 1

hsa-miR-578 1

hsa-miR-581 1

hsa-miR-587 1

hsa-miR-614

Although authors performed extensive experimentation downstream of hsa_circ_0007047, to further strengthen whether hsa_circ_0007047 directly targets miR-1178-3p, authors should perform mutation experiments by mutating miR-1178-3p binding sites in the hsa_circ_0007047 sequence.

AUUUCAGGCAAACAUGAGCAAG

|||||

GAUCCCUUCUUGUCACUCGUU

In the figure-8C, authors should include the sham group, along with MI, Over-circExo, and Shcirc-Exo.

Tube formation images are challenging to interpret, and please include higher magnification images.

Authors are suggested to include echocardiography all parameters in a table format and include as a supplementary table.

Minor comments

Authors are suggested to proofread for typos (For Eg: line 175 we fund)

Please include a scale bar for electron microscopy images.

Please include cognate gene names, e.g., circAMPK instead of has-circ nomenclature

Reviewer #2 (Remarks to the Author):

"Exosomal hsa_circ_0007047 attenuates post-myocardial infarction remodeling by promoting angiogenesis via miR-1178-3p/PDPK1 axis" is a manuscript aiming to reveal how exosomal circRNAs are regulated during the development of cardiovascular disease. The level of circRNAs is measured from the serum exosomes of myocardial infarction patients with or without cardiac remodeling and healthy controls and one circRNA, hsa_circ_0007047, is particularly identified and focused on further. This circRNA was studied in vitro using gain and loss of function studies. The same functional studies are carried out for miRNA, found to interact with circRNA and again for third molecule, supposedly a gene targeted by the miRNA. These three molecules are suggested to form a functional chain leading to the regulation of angiogenesis, cell death, proliferation, and migration. Furthermore, injection of circRNA overexpression vector to mice attenuated the myocardial infarction (MI) induced injury.

While apparently newly identified circRNA hsa_circ_0007047 is proposed in the manuscript, making it novel and interesting, the study lacks evidence to support all its claims and has concerning lack of precision in describing the results and stating the conclusions. The reader gets also easily confused by the rationale behind the study design and the decisions made, e.g. to analyze exosomal circRNAs from the serum while other evidence is studied in the cells or tissues.

To the reader, the focus of the manuscript appears to be on the functions of hsa_circ_0007047 and its molecular partners and only the therapeutic efficacy of hsa_circ_0007047 was studied in vivo. Thus, the stated expectations of the authors are not met as the regulation of hsa_circ_0007047 was not studied in MI (rows 160-161: "We anticipated that the obtained results would improve our understanding of exosomal circRNAs regulation during development of cardiovascular diseases").

Below are highlighted the major concerns of the reviewer in detail.

Q1 Claiming to study exosomes or EVs is not justified by the methods and characterization of exosome preparations

The first major concern is that already in the title, it is highlighted that the study is focused on exosomes and gives the reader the impression that the MI-altered exosome release or function, and the level of circRNA in exosomes, would be the key issues discussed in the study.

The International Society for Extracellular Vesicles maintains guidelines for reporting in extracellular vesicle studies. The terminology used for exosomes should be carefully reconsidered. In the light of the current knowledge, there is no definite way to distinguish or isolate specifically exosomes from other extracellular vesicle subtypes, such as microvesicles, as they overlap in size and have common molecular composition. Thus, it is recommended to use the term "extracellular vesicle" (EV) in this field. The authors should refer to the recently updated guidelines from International Society for Extracellular Vesicles (MISEV 2018).

The authors should carefully consider changing the whole manuscript to focus on the role of circRNA alone and leave out everything regarding exosomes or EVs. Currently, the methodology and characterization used in the paper (referring to "exosome isolation" and treatment with "exosomes harboring circRNA vector") is not appropriate to claim that the exosomes had any role in the seen effects. For the study to be of acceptable quality, it would need to be designed more precisely, considering which exosomes are being used (in the serum, the exosomes come from many different cells and major proportion from activated platelets), how, and how to validate that the exosomes are harboring the vectors (current method describes merely mixing and incubating exosomes with vectors).

The above details become difficult issues, since the precipitation method used for the isolation of exosomes is merely concentrating the initial sample (serum) and will contain lots of other proteins

(not part of exosomes, such as albumin), lipoproteins (having their own functions, not related to exosomes) and so on. Thus, while the study could claim biomarker applicability, or that the level of circRNAs was altered in precipitated serum between MI with and without remodeling, it is not correct to say it happened in exosomes based on this data.

The in vivo study, where exosomes harboring circRNA vectors were administered to the MI model mice, showed protective effects with overexpression vector. However, it is as likely (considering there is absolutely no validation of the association/incorporation etc. of the vectors to the exosomes) that the same effects could have been achieved with the vectors only. This is also an important control to include for these studies, as the vector alone should have less effect than the vector with the exosomes. In addition, there was no information on which exosomes were used for the in vivo experiment. Serum exosomes from the healthy controls? As mentioned above, serum exosomes obtained by commercial precipitation method suggest that these preparations also contained massive amounts of serum proteins in addition to the exosomes and vectors and it is left unclear what is the role of those.

Thus, for the science to be at an acceptable level regarding exosomes, the study design should be reformulated based on the MISEV 2018 guidelines and the selection of exosomes which are used for the in vivo part should be carefully considered. One very timely topic in the EV research is to answer a question from which cell types the functional EVs come from. Since the first part was detecting circRNA from serum, one good option would be to specifically enrich platelet derived EVs as they could be expected to have biological relevance in infarction as well. The other option for the study to be appropriate, would be to drop out all exosome claims.

Q2 Defective controls to claim involvement of miR-1178-3p/PDPK1 axis in the angiogenesis

The authors demonstrate the logical interaction between hsa_circ_0007047 and miR-1178-3p, between miR-1178-3p and PDPK1 and the effect of PDPK1 on the final functional outcome (angiogenesis etc.). However, both hsa_circ_0007047 and miR-1178-3p likely have other target genes in addition to PDPK1, as also pointed out by the authors. To claim involvement of this axis, another control experiment, where circRNA is overexpressed and the functional outcome is prevented by concomitant downregulation of PDPK1, would be needed.

The authors should justify focusing on only one of the putative miR-1178-3p targets based on the level of these genes in the serum alone. It is unclear why these were detected from the serum and what kind of information it provides to measure the mRNA and protein levels in the serum in this context, as it does not sound logical to the reviewer. These points should be clarified at least in the manuscript.

Q3 Unclear, partly non-scientific, language which can lead to misinterpretation of the meaning

Following sentences are examples which require revision:

1. Rows 248-250: "The results showed that there is a binding site of miR-1178-3p in the 3'UTR, suggesting that the expression of hsa_circ_0007047 might be regulated by miR-1178-3p"
2. Row 302: "To identify the downstream regulator of miR-1178-3p"
 - a. Incorrect meaning, probably should be downstream target of miR-1178-3p
 - b. Similar misuse of target/regulator term occurs several times in the manuscript and the authors should revise them to prevent confusion
3. Row 322 title: "miR-1178-3p-mediated PDPK1 regulate CMECs angiogenesis and VSMCs proliferation and migration"
4. Rows 349-350: "Altogether, these results indicated that miR-1178-3p-induced PDPK1 regulated the angiogenesis of CMECs and the proliferation and migration of VSMCs."

a. There are grammar issues, but it is also incorrect as miR-1178-3p does not induce the PDPK1

Q4 Other smaller comments

1. The authors should provide detailed description of the sequencing as it is currently missing. A reference should be provided if it was previously published.

2. Did the MI change the endogenous expression of the hsa_circ_0007047/miR-1178-3p/PDPK1 in the in vivo MI model, in relevant cell types/tissues? These should be reported as it would help the reasoning why serum level was analyzed in the first place and if it could have any functional implications for the cardiac remodeling.

3. Western blot is missing negative controls (such as whole serum) and detection of (contaminant) proteins that should not be in exosomes.

4. Rows 399-400: "In the present study, exosomal hsa_circ_0007047 was identified as a novel regulator of the postinfarct ventricular remodeling."

a. The authors should reformulate the claim as it gives an impression, that this circRNA was altered in the in vivo MI model in exosomes.

b. The authors should revise the claims throughout the manuscript as similar issues were repeating

5. The literature review in the introduction and discussion

a. The authors should clarify why so many previous studies are described in detail and what was the relevance for this manuscript

b. The authors should check the claims and references for those claims, e.g.

i. Rows 408-410: "It has been recently demonstrated that exosomes are considered as the main mediators of intercellular communication in the myocardium, which may attenuate the development of detrimental structural changes and the consequent heart failure after MI 20"

ii. The current literature is not sufficient to claim that the exosomes would be the main intercellular communication mediators in the heart (or anywhere else) as there are many other ways for the communication as well.

iii. The attenuation of MI damage would not be in line with the change the authors demonstrated in the serum as based on the presented hypothesis in this manuscript, the downregulation of exosomal hsa_circ_0007047 due to MI would promote harmful remodeling. The authors should clarify if this was misunderstanding.

c. Rows: 457-459: "miR-1178-3p not only binds to hsa_circ_0007047 but also to PDPK1, the leucyl-specific aminopeptidase activated factor, contributing to the VEGF-dependent activation of S6K"

i. The authors did not study the VEGF dependency in this context, the sentence should be revised or appropriately referenced.

Reviewer #3 (Remarks to the Author):

In this manuscript, the authors investigate the effects of circRNA, hsa_circ_0007047, in the heart and under myocardial infarction. Both in vitro and in vivo experiments are provided to understand the mechanism of action for this circRNA, which acts as miRNA sponge to sequester miR-1178-3p that binds PDPK1. The most crucial information about how the target circRNA was derived is not provided, which raises a significant suspicion about the aim of this study. More specific comments are listed below:

Major points:

[1] Line 167 "Our initial whole transcriptome sequencing analysis...": No information is provided regarding this RNA-seq data and analysis. The authors must deposit RNA-seq data in a public domain, such as Gene Expression Omnibus (GEO).

[2] Line 171 "hsa_circ_0000212, hsa_circ_0089282 and hsa_circ_0007047": The authors must provide the genomic coordinate of each circRNA along with the information about its host gene.

[3] Lines 352 - 356 "To further confirm the beneficial effects of the exosomal hsa_circ_0007047 in the postinfarct ventricular remodeling, exosomes containing overexpressing or suppressing hsa_circ_0007047 plasmid were injected into the tail vein of mice after induction of their experimental MI." Given that exosomes were injected into the tail vein of mice, the authors must provide the detailed analysis of the effects of this injection to other major organs/tissues, such as liver, lung, and muscle.

[4] Figure 9C. Based on the images provided, the protein expression of PDPK1 is several folds higher in the Over-circ-Exo samples compared to the control. The authors must quantify such changes by performing Western blotting and ELISA assays.

[5] Which cell type(s) express hsa_circ_0007047 and miR-1178-3p.

Minor points:

(1) Line 303 "TargetScan7.2, miRwalk and miRDB". These bioinformatic tools must be cited with their original publications.

(2) One- or two-tail t-test?

(3) The sample size for each result is missing in the Figure Legends.

(4) In Western blotting results, molecular markers are missing. Furthermore, the authors must provide more than $n = 1$ per condition on the same SDS-PAGE gel. It is now a common practice to provide the image of whole membrane as supplementary data, which the authors must provide such data in this manuscript.

Reviewers' comments:

Reviewer #1 (Remarks to the Author):

I would recommend a substantial revision addressing my following concerns
The current manuscript by Shuai et al., titled “Exosomal hsa_circ_0007047 attenuates post-myocardial infarction remodeling by promoting angiogenesis via miR-1178-3p/PDPK1 axis” extensive experimentation and suggest that hsa_circ_0007047 promoted angiogenesis both in vitro and in vivo. Importantly overexpression of hsa_circ_0007047 in mice significantly improved fibrosis and cardiac function post-MI in mouse models. However, I have the following concerns.

Major concerns

How efficient was the plasmid able to include/reduce hsa-circ_0007047 expression in the mouse? And for how long

Reply: Thanks for your comments. We have detected the expression of hsa-circ_0007047 (circCEBPZOS) at 28 days after surgical procedures for MI model. The results showed that the expression of circCEBPZOS was significantly downregulated compared with sham group. However, the expression of circCEBPZOS was successfully overexpressed or suppressed when treatment with exosomes containing overexpressing or suppressing circCEBPZOS plasmid in the MI mice (**Figure 8 A**).

Figure 8. circCEBPZOS alleviated postinfarct cardiac remodeling

A, qRT-PCR was used to verify the expression of circCEBPZOS in the heart tissue treatment with exosomal circCEBPZOS, GAPDH served as control.

Authors should demonstrate which cells in the heart takes up these plasmids?

Reply: Thanks very much. The expression of circCEBPZOS and miR-1178-3p was detected by qRT-PCR in the cardiomyocytes, fibroblasts and cardiac microvascular endothelial cells, respectively. We found that both circCEBPZOS and miR-1178-3p was prefer to express in cardiac microvascular endothelial cells (Shown in **Figure S**).

Figure S. The expression of circCEBPZOS and miR-1178-3p detected by qRT-PCR in the cardiomyocytes, fibroblasts and cardiac microvascular endothelial cells, respectively.

Does a human ortholog of hsa_circ_0007047 exist in mice? If yes, what happens to its levels post-MI mice?

Reply: Thanks for your comments. We have performed Sequence Alignment by Blast in NCBI. The results showed that there was no human ortholog of hsa_circ_0007047 exist in mice.

Description	Scientific Name	Max Score	Total Score	Query Cover	E value	Per Ident	Acc. Len	Accession
PREDICTED_Pan paniscus CEBPZ opposite strand (CEBPZOS) transcript variant X3 mRNA	Pan paniscus	455	455	100%	9e-124	100.00%	313	XM_024952950.1
PREDICTED_Pan paniscus CEBPZ opposite strand (CEBPZOS) transcript variant X2 mRNA	Pan paniscus	455	455	100%	9e-124	100.00%	1497	XM_024927594.2
PREDICTED_Pan paniscus CEBPZ opposite strand (CEBPZOS) transcript variant X1 mRNA	Pan paniscus	455	455	100%	9e-124	100.00%	979	XM_024927503.2
Homo sapiens CEBPZ opposite strand (CEBPZOS) transcript variant 5 non-coding RNA	Homo sapiens	455	455	100%	9e-124	100.00%	865	NR_136316.2
Homo sapiens CEBPZ opposite strand (CEBPZOS) transcript variant 1 mRNA	Homo sapiens	455	455	100%	9e-124	100.00%	3331	NM_001322373.2
Homo sapiens CEBPZ opposite strand (CEBPZOS) transcript variant 4 mRNA	Homo sapiens	455	455	100%	9e-124	100.00%	711	NM_001322375.2
Homo sapiens CEBPZ opposite strand (CEBPZOS) transcript variant 3 mRNA	Homo sapiens	455	455	100%	9e-124	100.00%	3148	NM_001322376.2
Homo sapiens CEBPZ opposite strand (CEBPZOS) transcript variant 2 mRNA	Homo sapiens	455	455	100%	9e-124	100.00%	3152	NM_001322374.2
PREDICTED_Homo sapiens CEBPZ opposite strand (CEBPZOS) transcript variant X4 mRNA	Homo sapiens	455	455	100%	9e-124	100.00%	293	XM_017003109.2
PREDICTED_Homo sapiens CEBPZ opposite strand (CEBPZOS) transcript variant X3 mRNA	Homo sapiens	455	455	100%	9e-124	100.00%	531	XM_017003108.2
PREDICTED_Homo sapiens CEBPZ opposite strand (CEBPZOS) transcript variant X2 mRNA	Homo sapiens	455	455	100%	9e-124	100.00%	3216	XM_017003107.1
PREDICTED_Homo sapiens CEBPZ opposite strand (CEBPZOS) transcript variant X1 mRNA	Homo sapiens	455	455	100%	9e-124	100.00%	3395	XM_017003106.1
Homo sapiens cDNA FL110952 fl. clone PLACE109374	Homo sapiens	455	455	100%	9e-124	100.00%	1871	AK091814.1
Homo sapiens clone IMAGE388733 mRNA	Homo sapiens	455	455	100%	9e-124	100.00%	1251	BC017652.1
PREDICTED_Hylobates moloch CEBPZ opposite strand (CEBPZOS) transcript variant X3 mRNA	Hylobates moloch	449	449	100%	4e-122	99.59%	723	XM_032178274.1
PREDICTED_Hylobates moloch CEBPZ opposite strand (CEBPZOS) transcript variant X2 mRNA	Hylobates moloch	449	449	100%	4e-122	99.59%	1073	XM_032178273.1
PREDICTED_Hylobates moloch CEBPZ opposite strand (CEBPZOS) transcript variant X1 mRNA	Hylobates moloch	449	449	100%	4e-122	99.59%	604	XM_032178272.1
PREDICTED_Gorilla gorilla gorilla CEBPZ opposite strand (CEBPZOS) transcript variant X5 mRNA	Gorilla gorilla gorilla	449	449	100%	4e-122	99.59%	716	XM_031008619.1
PREDICTED_Gorilla gorilla gorilla CEBPZ opposite strand (CEBPZOS) transcript variant X4 mRNA	Gorilla gorilla gorilla	449	449	100%	4e-122	99.59%	598	XM_019021204.2
PREDICTED_Gorilla gorilla gorilla CEBPZ opposite strand (CEBPZOS) transcript variant X3 mRNA	Gorilla gorilla gorilla	449	449	100%	4e-122	99.59%	777	XM_019021203.2
PREDICTED_Gorilla gorilla gorilla CEBPZ opposite strand (CEBPZOS) transcript variant X2 mRNA	Gorilla gorilla gorilla	449	449	100%	4e-122	99.59%	799	XM_031008612.1
PREDICTED_Gorilla gorilla gorilla CEBPZ opposite strand (CEBPZOS) transcript variant X1 mRNA	Gorilla gorilla gorilla	449	449	100%	4e-122	99.59%	942	XM_019021202.2
PREDICTED_Nomascus leucogenys CEBPZ opposite strand (CEBPZOS) transcript variant X2 mRNA	Nomascus leuc-	449	449	100%	4e-122	99.59%	716	XM_030800568.1
PREDICTED_Nomascus leucogenys CEBPZ opposite strand (CEBPZOS) transcript variant X1 mRNA	Nomascus leuc-	449	449	100%	4e-122	99.59%	598	XM_030800567.1
PREDICTED_Pan troglodytes CEBPZ opposite strand (CEBPZOS) transcript variant X4 mRNA	Pan troglodytes	449	449	100%	4e-122	99.59%	412	XM_016948370.2
PREDICTED_Pan troglodytes CEBPZ opposite strand (CEBPZOS) transcript variant X3 mRNA	Pan troglodytes	449	449	100%	4e-122	99.59%	1286	XM_016948369.2
PREDICTED_Pan troglodytes CEBPZ opposite strand (CEBPZOS) transcript variant X2 mRNA	Pan troglodytes	449	449	100%	4e-122	99.59%	1609	XM_016948368.2

Does over-expression plasmids for hsa-circ_0007047 affect miR-1178 levels in the heart post-MI? Also, in general, what are miR-1178 levels post-MI in mouse hearts without hsa-circ_0007047 treatment.

Reply: Thanks for your comments. The miR-1178-3p levels was upregulated in mouse hearts with MI. However, the expression of miR-1178-3p could be reduced when overexpressing CircCEBPZOS while increased when suppressing CircCEBPZOS expression compared than that in the MI mouse (**Shown in Figure 9A**).

Figure 9. The expression changes of miR-1178-3p and PDPK1 in vivo treatment with exosomes harboring circCEBPZOS. A, The expression of miR-1178-3p was detected by qRT-PCR, U6 act as control;

hsa_circ_0007047 has how many binding sites for miR-1178-3p; authors should mention this at some point. Circinteractome shows hsa_circ_0007047 has binding sites for the following miRs, what was the rationale in choosing miR-1178? hsa-miR-11781 hsa-miR-12901 hsa-miR-548k1 hsa-miR-5781 hsa-miR-5811 hsa-miR-5871 hsa-miR-614

Reply: Thanks for your comments. The binding sites of miR-1178-3p to hsa_circ_0007047 was predicted by Circinteractome (<https://circinteractome.nia.nih.gov/>). The results showed that there was only one potential binding site between miR-1178-3p and hsa_circ_0007047.

TargetScan miRNA predictions												
CircRNA Mirbase ID	CircRNA (Top) - miRNA (Bottom) pairing	Site Type	CircRNA Start	CircRNA End	3' pairing	local AU	position	TA	SPS	context+ score	context+ score percentile	
hsa_circ_0007047 (5' - 3')	AATTGAGGCAACAACTAGGCAAG (11111) GAGCCGCGCCGCGCCACCCGCU	7mer-1a	132	138	0.001	-0.006	-0.038	-0.001	-0.032	-0.15	89	
hsa_circ_0007047 (5' - 3')	UUGAAGGUGUUGUUCAGAGCCGC (11111) AGGCACTAGGCGUUCAGGU	7mer-1a	170	176	0.004	-0.013	-0.042	0.010	0.038	-0.077	91	
hsa_circ_0007047 (5' - 3')	GGGGHHHGGGAGGCGCCGCGCCGCU (11111) GCGCUGUAGGCGCGCCAGHGG	7mer-m8	9	15	too_close	too_close	too_close	too_close	too_close	too_close	NA	
hsa_circ_0007047 (5' - 3')	CAGCCAGCAGUAGCAGAGAAU (11111) UGGGAGGAGCGCGCGCGCGCU	7mer-1a	136	142	0.004	-0.017	-0.039	0.017	0.042	-0.067	86	
hsa_circ_0007047 (5' - 3')	UGUUGAGCGAGGAGCAGCAGAGC (11111) UGACAGAGCGCGCGCGCGCU	7mer-m8	105	111	0.003	0.029	-0.047	-0.011	0.020	-0.126	85	
hsa_circ_0007047 (5' - 3')	GAGAAAGUGGAGAGGAGAGGAGAG (11111) CAGCGAGAGGAGGAGAGCGCU	7mer-m8	194	200	0.003	0.001	-0.055	0.023	0.043	-0.105	93	
hsa_circ_0007047 (5' - 3')	GGGAGCGGAGCGCGCGCGCGCU (11111) GGGAGCGCGCGCGCGCGAG	7mer-m8	67	73	0.003	-0.055	-0.052	-0.097	-0.027	-0.348	96	

In addition, the results showed that miR-614, miR-587, miR-1290 and miR-1178-3p have the highest score Context+ Score which could bind with there is a binding site of miR-1178-3p in the 3'UTR. Further analysis showed that the expression of miR-1178-3p was significantly elevated in the exosomes isolated from serum of patients with postinfarct cardiac remodeling, when compared with non-postinfarct cardiac remodeling and control group. However, the expression of miR-614 and miR-587 was significantly decreased in the exosomes isolated from serum of patients with postinfarct cardiac remodeling, when compared with non-postinfarct cardiac remodeling and

control group. The expression of miR-1290 was dramatically upregulated in the non-postinfarct cardiac remodeling group while not changed in the postinfarct cardiac remodeling. Therefore, miR-1178-3p was used for further study. The new evidence has been added in the revised manuscript shown in **Figure 4A and B**.

Although authors performed extensive experimentation downstream of hsa_circ_0007047, to further strengthen whether hsa_circ_0007047 directly targets miR-1178-3p, authors should perform mutation experiments by mutating miR-1178-3p binding sites in the hsa_circ_0007047 sequence.

AUUUCAGGCAAACAAUGAGCAAG

|||||

GAUCCCUUCUUGUCACUCGUU

Reply: Thanks for your comments. The mutation experiment has been performed with binding sites in the hsa_circ_0007047 sequence mutation which was shown in Figure 4C.

In the figure-8C, authors should include the sham group, along with MI, Over-circExo, and Shcirc-Exo.

Reply: Thanks for your comments. Sham group has been added in Figure 8C.

Tube formation images are challenging to interpret, and please include higher magnification images.

Reply: Thanks for your comments. The tube formation images in Figure 3B, Figure 5B and Figure 7C have been changed to higher magnification images.

Authors are suggested to include echocardiography all parameters in a table format and include as a supplementary table.

Reply: Thanks for your comments. Echocardiography all parameters has been added in the manuscript in a table format as supplementary table 3.

Supplementary Table 3. Echocardiographic characteristics.

	Sham	MI	Over-circ-Exo	Sh-circ-Exo
LVEDD (mm)	3.99±0.12	4.48±0.31	3.94±0.21**	4.69±0.35###

| | | | | |
|---|---|---|---|---|
| **LVESD (mm)** | 3.05±0.13 | 3.16±0.23 | 2.99±0.33** | 3.24±0.26## |
| **LVEF %** | 47.65±3.44 | 41.65±5.27 | 55.25±4.53** | 35.67±5.34## |
| **LVFS %** | 23.58±2.31 | 20.18±4.35 | 24.48±4.33** | 20.02±5.09## |
| **SV(μL)** | 33.29±4.19 | 31.48±6.16 | 34.69±5.31 | 30.54±6.19 |

LVEDD, left ventricular end-diastolic diameter; LVESD, left ventricular end-systolic diameter; FS, fractional shortening; and EF, ejection fraction. All values are mean±SE.

**Minor comments**

**Authors are suggested to proofread for typos (For Eg: line 175 we fund)**

***Reply:*** Thanks for your comments. We have proofread for the full manuscript by a professional English editing agency.

Certificate of Elsevier
Language Editing Services

The following article was edited by Elsevier Language Editing Services:
"Extracellular vesicles-derived circCEBPZOS attenuates postinfarct remodeling by promoting angiogenesis via miR-1178-3p/PDPK1 axis"

Authored by:
Shuai MAO

Date: 16-Jan-2022
Serial number: LE-230553-63C528105B95

**Please include a scale bar for electron microscopy images.**

***Reply:*** Thanks for your comments. The scale bar for electron microscopy images was 100μm which exist in the bottom right corner of the images. In order to clearer display the scale bar for electron microscopy images, we relabeled the scale bar in **Figure 1D**.

**Please include cognate gene names, e.g., circAMPK instead of has-circ nomenclature**

***Reply:*** Thanks for your comments. We have changed has_circ_0007047 to CircCEBPZOS according to the annotation from circbase (http://www.circbase.org/) and NCBI.

CEBPZOS CEBPZ opposite strand [*Homo sapiens* (human)]

Gene ID: 105905675, updated on 11-Jun-2021

Summary

Official Symbol	CEBPZOS <small>provided by HGNC</small>
Official Full Name	CEBPZ opposite strand <small>provided by HGNC</small>
Primary source	HGNC,HGNC:49288
See related	Ensembl:ENSG00000216739
Gene type	protein coding
RefSeq status	VALIDATED
Organism	Homo sapiens
Lineage	Eukaryota; Metazoa; Chordata; Cranata; Vertebrata; Euteleostomi; Mammalia; Eutheria; Euarchontoglires; Primates; Haplorhini; Catarrhini; Hominoidea; Homo
Also known as	CEBPZ-AS1
Expression	Ubiquitous expression in adrenal (RPKM 10.4), thyroid (RPKM 9.4) and 25 other tissues See more
Orthologs	mouse af
	 Try the new Gene table
	Try the new Transcript table

Reviewer #2 (Remarks to the Author):

“Exosomal hsa_circ_0007047 attenuates post-myocardial infarction remodeling by promoting angiogenesis via miR-1178-3p/PDPK1 axis” is a manuscript aiming to reveal how exosomal circRNAs are regulated during the development of cardiovascular disease. The level of circRNAs is measured from the serum exosomes of myocardial infarction patients with or without cardiac remodeling and healthy controls and one circRNA, hsa_circ_0007047, is particularly identified and focused on further. This circRNA was studied *in vitro* using gain and loss of function studies. The same functional studies are carried out for miRNA, found to interact with circRNA and again for third molecule, supposedly a gene targeted by the miRNA. These three molecules are suggested to form a functional chain leading to the regulation of angiogenesis, cell death, proliferation, and migration. Furthermore, injection of circRNA overexpression vector to mice attenuated the myocardial infarction (MI) induced injury.

While apparently newly identified circRNA hsa_circ_0007047 is proposed in the manuscript, making it novel and interesting, the study lacks evidence to support all its claims and has concerning lack of precision in describing the results and stating the conclusions. The reader gets also easily confused by the rationale behind the study design and the decisions made, e.g. to analyze exosomal circRNAs from the serum while other evidence is studied in the cells or tissues.

To the reader, the focus of the manuscript appears to be on the functions of hsa_circ_0007047 and its molecular partners and only the therapeutic efficacy of hsa_circ_0007047 was studied *in vivo*. Thus, the stated expectations of the authors are not met as the regulation of hsa_circ_0007047 was not studied in MI (rows 160-161: “We anticipated that the obtained results would improve our understanding of exosomal circRNAs regulation during development of cardiovascular diseases”).

Reply: Thanks for your comments. We compared the expression profiles of EVs-derived circRNAs in patients with and without cardiac remodeling using a high-throughput RNA sequencing (Accession number GSE194388). First, we tested function of EVs-derived circCEBPZOS, which has already been related to LV remodeling, *in vitro* and *in vivo* using silencing and overexpression strategies. Then, we explored the mechanisms underlying the actions of circCEBPZOS during the LV remodeling after MI. We anticipated that the obtained results

would improve our understanding of EVs-derived circCEBPZOS regulation during development of post-infarct cardiac remodeling.

Below are highlighted the major concerns of the reviewer in detail.

Q1 Claiming to study exosomes or EVs is not justified by the methods and characterization of exosome preparations.

The first major concern is that already in the title, it is highlighted that the study is focused on exosomes and gives the reader the impression that the MI-altered exosome release or function, and the level of circRNA in exosomes, would be the key issues discussed in the study.

Reply: Thanks for your comments. The study focused on the function and mechanism of circRNA from exosomes with MI. Whether the exosome release could be affected by MI will be explored in the further study.

The International Society for Extracellular Vesicles maintains guidelines for reporting in extracellular vesicle studies. The terminology used for exosomes should be carefully reconsidered. In the light of the current knowledge, there is no definite way to distinguish or isolate specifically exosomes from other extracellular vesicle subtypes, such as microvesicles, as they overlap in size and have common molecular composition. Thus, it is recommended to use the term “extracellular vesicle” (EV) in this field. The authors should refer to the recently updated guidelines from International Society for Extracellular Vesicles (MISEV 2018).

Reply: Thanks for your comments. We have changed exosomes to “extracellular vesicle” (EV) in the manuscript. Thus, the new title of the manuscript is *Extracellular vesicle-derived circCEBPZOS attenuates pos-myocardial infarction remodeling by promoting angiogenesis via miR-1178-3p/PDPK1 axis.*

The authors should carefully consider changing the whole manuscript to focus on the role of circRNA alone and leave out everything regarding exosomes or EVs. Currently, the methodology and characterization used in the paper (referring to “exosome isolation” and treatment with “exosomes harboring circRNA vector”) is not appropriate to claim that the

exosomes had any role in the seen effects. For the study to be of acceptable quality, it would need to be designed more precisely, considering which exosomes are being used (in the serum, the exosomes come from many different cells and major proportion from activated platelets), how, and how to validate that the exosomes are harboring the vectors (current method describes merely mixing and incubating exosomes with vectors).

The above details become difficult issues, since the precipitation method used for the isolation of exosomes is merely concentrating the initial sample (serum) and will contain lots of other proteins (not part of exosomes, such as albumin), lipoproteins (having their own functions, not related to exosomes) and so on. Thus, while the study could claim biomarker applicability, or that the level of circRNAs was altered in precipitated serum between MI with and without remodeling, it is not correct to say it happened in exosomes based on this data.

Reply: Thanks for your comments. We totally agree with your comments. It is better to confirm the exosomes from which cells that used for more precisely study. However, it is difficult to obtain clinical samples of heart tissue, so it cannot be verified in clinical tissues and we choose serum to perform the study. In addition, we have set up a MI-NC group besides sham group which could eliminate some of the bad effects caused by other proteins or lipoproteins.

The *in vivo* study, where exosomes harboring circRNA vectors were administered to the MI model mice, showed protective effects with overexpression vector. However, it is as likely (considering there is absolutely no validation of the association/incorporation etc. of the vectors to the exosomes) that the same effects could have been achieved with the vectors only. This is also an important control to include for these studies, as the vector alone should have less effect than the vector with the exosomes. In addition, there was no information on which exosomes were used for the *in vivo* experiment. Serum exosomes from the healthy controls? As mentioned above, serum exosomes obtained by commercial precipitation method suggest that these preparations also contained massive amounts of serum proteins in addition to the exosomes and vectors and it is left unclear what is the role of those.

Reply: Thanks for your comments. We have performed the function of hsa_circ_0007047 alone in MI model mouse. In fact, the similar effects could be achieved transfection with the vectors without exosomes to MI mouse (**Supplementary Figure 4**). However, in terms of the key

circRNAs was excavated by whole transcriptome sequencing analysis in exosomes derived from sera of patients with and without postinfarct cardiac remodeling. Therefore, we must focus the study on exosomes which carried circRNA hsa_circ_0007047. In addition, we have set up a MI-NC group besides sham group which could eliminate some of the bad effects caused by other proteins or lipoproteins.

Thus, for the science to be at an acceptable level regarding exosomes, the study design should be reformulated based on the MISEV 2018 guidelines and the selection of exosomes which are used for the in vivo part should be carefully considered. One very timely topic in the EV research is to answer a question from which cell types the functional EVs come from. Since the first part was detecting circRNA from serum, one good option would be to specifically enrich platelet derived EVs as they could be expected to have biological relevance in infarction as well. The other option for the study to be appropriate, would be to drop out all exosome claims.

Reply:

Thanks for your comments. We will focus the study on the platelet derived EVs as they could be expected to have biological relevance in infarction according to your comments in the future study.

Q2 Defective controls to claim involvement of miR-1178-3p/PDPK1 axis in the angiogenesis
The authors demonstrate the logical interaction between hsa_circ_0007047 and miR-1178-3p, between miR-1178-3p and PDPK1 and the effect of PDPK1 on the final functional outcome (angiogenesis etc.). However, both hsa_circ_0007047 and miR-1178-3p likely have other target genes in addition to PDPK1, as also pointed out by the authors. To claim involvement of this axis, another control experiment, where circRNA is overexpressed and the functional outcome is prevented by concomitant downregulation of PDPK1, would be needed.

Reply: Thanks for your comments. The recovery experiment for silence of PDPK1 in the overexpressing hsa_circ_0007047 was performed. We found that the function of overexpressing circCEBPZOS on the CMECs angiogenesis and VSMCs proliferation and migration could be reversed by administration with sh-PDPK1 (**Supplementary Figure 2**).

The authors should justify focusing on only one of the putative miR-1178-3p targets based on the level of these genes in the serum alone. It is unclear why these were detected from the

serum and what kind of information it provides to measure the mRNA and protein levels in the serum in this context, as it does not sound logical to the reviewer. These points should be clarified at least in the manuscript.

Reply: Thanks for your comments. It is difficult to obtain clinical samples of heart tissue, so it cannot be verified in clinical tissues and we choose serum to perform the study. In addition, the potential binding miRNA targets of hsa_circ_0007047 was predicted by Circinteractome (<https://circinteractome.nia.nih.gov/>). And the results showed that miR-614, miR-587, miR-1290 and miR-1178-3p have the highest score Context+ Score which could bind with there is a binding site of miR-1178-3p in the 3'UTR (**Figure 4A**). Further analysis showed that the expression of miR-1178-3p was significantly elevated in the exosomes isolated from serum of patients with postinfarct cardiac remodeling, when compared with non-postinfarct cardiac remodeling and control group. However, the expression of miR-614 and miR-587 was significantly decreased in the exosomes isolated from serum of patients with postinfarct cardiac remodeling, when compared with non-postinfarct cardiac remodeling and control group. The expression of miR-1290 was dramatically upregulated in the non-postinfarct cardiac remodeling group while not changed in the postinfarct cardiac remodeling (**Figure 4B**). Therefore, miR-1178-3p was used for further study. The new evidence has been added in the revised manuscript. In addition, we also detect the expression of miR-1178-3p and PDPK1 in the MI mouse. The results indicated that the expression of miR-1178-3p levels was upregulated while the expression of PDPK1 was downregulated in mouse hearts with MI (**Figure 9A and 9B**).

TargetScan miRNA predictions												
CircRNA Mirbase ID	CircRNA (Top) - miRNA (Bottom) pairing	Site Type	CircRNA Start	CircRNA End	3' pairing	local AU	position	TA	SPS	context+ score	context+ score percentile	
hsa_circ_0007047 (3' - 5')	<pre> ATTTCCGGCCAAACAAAGGCGAAG GATCCUUCUUCUUCUUCUUCUUCUUC </pre>	7mer-1a	132	138	0.001	-0.006	-0.038	-0.001	-0.032	-0.15	89	
hsa_circ_0007047 (5' - 3')	<pre> UUGAAGUUUUAUUCUCAAUCCAC AGGACUAGUUUUUUUAGUU </pre>	7mer-1a	170	176	0.004	-0.013	-0.042	0.010	0.038	-0.077	91	
hsa_circ_0007047 (5' - 3')	<pre> #H#H#H#G#A#S#C#C#C#C#C#C#U#U# C#C#U#U#A#G#C#C#U#C#A#U#A#A#A# </pre>	7mer-m8	9	15	too_close	too_close	too_close	too_close	too_close	too_close	NA	
hsa_circ_0007047 (5' - 3')	<pre> CAGGCAACCAUUAACCGAGAAAT CUCUGAGAUUCUUUUUCUUC </pre>	7mer-1a	136	142	0.004	-0.017	-0.039	0.017	0.042	-0.067	86	
hsa_circ_0007047 (5' - 3')	<pre> UUUUUACAAAGUUCACAGAAAC UGACUAGAUUCUUUUUCUUC </pre>	7mer-m8	105	111	0.003	0.029	-0.047	-0.011	0.020	-0.126	89	
hsa_circ_0007047 (5' - 3')	<pre> G#A#A#U#U#U#A#A#U#C#A#U#A#A# C#C#U#U#A#G#C#C#U#C#A#U#A#A# </pre>	7mer-m8	194	200	0.003	0.021	-0.055	0.023	0.043	-0.105	93	
hsa_circ_0007047 (5' - 3')	<pre> GUAGCCGAAACUUUAGCCUUU G#U#A#C#C#U#U#U#U#U#C#A#A# </pre>	7mer-m8	67	73	0.003	-0.055	-0.052	-0.097	-0.027	-0.348	96	
hsa_circ_0007047 (3' - 5')												

Q3 Unclear, partly non-scientific, language which can lead to misinterpretation of the meaning

Following sentences are examples which require revision:

- 1. Rows 248-250: “The results showed that there is a binding site of miR-1178-3p in the 3’UTR, suggesting that the expression of hsa_circ_0007047 might be regulated by miR-1178-3p”**
- 2. Row 302: “To identify the downstream regulator of miR-1178-3p”**
 - a. Incorrect meaning, probably should be downstream target of miR-1178-3p**
 - b. Similar misuse of target/regulator term occurs several times in the manuscript and the authors should revise them to prevent confusion**
- 3. Row 322 title: “miR-1178-3p-mediated PDPK1 regulate CMECs angiogenesis and VSMCs proliferation and migration”**
- 4. Rows 349-350: “Altogether, these results indicated that miR-1178-3p-induced PDPK1 regulated the angiogenesis of CMECs and the proliferation and migration of VSMCs.”**
 - a. There are grammar issues, but it is also incorrect as miR-1178-3p does not induce the PDPK1**

Reply: Thanks for your comments. We have proofread for the full manuscript by a professional English editing agency.

Q4 Other smaller comments

1. The authors should provide detailed description of the sequencing as it is currently missing. A reference should be provided if it was previously published.

Reply: Thanks for your comments. The RNA sequencing data have been deposited in NCBI's Gene Expression Omnibus and are accessible through GEO Series accession number GSE194388. The following secure token has been created to allow review of record GSE194388 while it remains in private status: ypcnucikllqvtgz.

2. Did the MI change the endogenous expression of the hsa_circ_0007047/miR-1178-3p/PDPK1 in the in vivo MI model, in relevant cell types/tissues? These should be reported as it would help the reasoning why serum level was analyzed in the first place and if it could have any functional implications for the cardiac remodeling.

Reply: Thanks for your comments. The expression of hsa_circ_0007047, miR-1178-3p and PDPK1 was measured in the MI mouse. The results indicated that the expression of miR-1178-3p and PDPK1 in the MI mouse. The results indicated that the expression of miR-1178-3p levels was upregulated (**Figure 9A**) while the expression of hsa_circ_0007047 and PDPK1 was downregulated in mouse hearts with MI (**Figure 8A and 9B**).

3. Western blot is missing negative controls (such as whole serum) and detection of (contaminant) proteins that should not be in exosomes.

Reply:

Thanks for your comments. It is more accurate to detect the exosomes with negative controls or proteins that not in exosomes. However, it is generally sufficient to verify only the markers of exosomes and we have added the references in the manuscript.

1) *Exosomes isolated from the plasma of remote ischemic conditioning rats improved cardiac function and angiogenesis after myocardial infarction through targeting Hsp70*, *Aging (Albany NY)*. 2020 Feb 29; 12(4): 3682–3693.

2) *Exosomal lncRNA AK139128 Derived from Hypoxic Cardiomyocytes Promotes Apoptosis and Inhibits Cell Proliferation in Cardiac Fibroblasts*. *Int J Nanomedicine*. 2020; 15: 3363–3376.

4. Rows 399-400: “In the present study, exosomal hsa_circ_0007047 was identified as a novel regulator of the postinfarct ventricular remodeling.”

a. The authors should reformulate the claim as it gives an impression, that this circRNA was altered in the in vivo MI model in exosomes.

b. The authors should revise the claims throughout the manuscript as similar issues were repeating

Reply:

Thanks for your comments. We have revised the claims throughout the manuscript.

5. The literature review in the introduction and discussion

a. The authors should clarify why so many previous studies are described in detail and what was the relevance for this manuscript

b. The authors should check the claims and references for those claims, e.g.

i. Rows 408-410: “It has been recently demonstrated that exosomes are considered as the main mediators of intercellular communication in the myocardium, which may attenuate the development of detrimental structural changes and the consequent heart failure after MI 20”

ii. The current literature is not sufficient to claim that the exosomes would be the main intercellular communication mediators in the heart (or anywhere else)

Reply: Thanks for your comments. We have revised the claims throughout the manuscript.

as there are many other ways for the communication as well.

iii. The attenuation of MI damage would not be in line with the change the authors demonstrated in the serum as based on the presented hypothesis in this manuscript, the downregulation of exosomal hsa_circ_0007047 due to MI would promote harmful remodeling. The authors should clarify if this was misunderstanding.

Reply: Thanks for your comments. The expression of hsa_circ_0007047 was downregulated in the exosomes from MI, suggesting that downregulating exosomal hsa_circ_0007047 contribute to MI.

c. Rows: 457-459: “miR-1178-3p not only binds to hsa_circ_0007047 but also to PDPK1, the leucyl-specific aminopeptidase activated factor, contributing to the VEGF-dependent activation of S6K”

i. The authors did not study the VEGF dependency in this context, the sentence should be revised or appropriately referenced.

Reply:

Thanks for your comments. The reference 35 indicated that PDPK1 is a leucyl-specific aminopeptidase activated factor which contribute to the VEGF-dependent activation of S6K.

Reviewer #3 (Remarks to the Author):

In this manuscript, the authors investigate the effects of circRNA, hsa_circ_0007047, in the heart and under myocardial infarction. Both in vitro and in vivo experiments are provided to understand the mechanism of action for this circRNA, which acts as miRNA sponge to sequester miR-1178-3p that binds PDPK1.

The most crucial information about how the target circRNA was derived is not provided, which raises a significant suspicion about the aim of this study.

The RNA sequencing data have been deposited in NCBI's Gene Expression Omnibus and are accessible through GEO Series accession number GSE194388. The following secure token has been created to allow review of record GSE194388 while it remains in private status: ypcnucikllqvtgz.

More specific comments are listed below:

Major points:

[1] Line 167 “Our initial whole transcriptome sequencing analysis...”: No information is provided regarding this RNA-seq data and analysis. The authors must deposit RNA-seq data in a public domain, such as Gene Expression Omnibus (GEO).

Reply: Thanks for your comments. The detailed description of the sequencing has been existed in our another paper which has not published. The RNA sequencing data have been deposited in NCBI's Gene Expression Omnibus and are accessible through GEO Series accession number GSE194388. The following secure token has been created to allow review of record GSE194388 while it remains in private status: ypcnucikllqvtgz.

[2] Line 171 “hsa_circ_0000212, hsa_circ_0089282 and hsa_circ_0007047”: The authors must provide the genomic coordinate of each circRNA along with the information about its host gene.

Reply: Thanks for your comments. The genomic coordinate of each circRNA has been provided in the manuscript according to the circBase. The hsa_circ_0000212, hsa_circ_0089282 and hsa_circ_0007047 was localized on chr10:7405839-7423911, chr9:135172273-135187243 and chr2:37426846-37428869, respectively.

[3] Lines 352 - 356 "To further confirm the beneficial effects of the exosomal hsa_circ_0007047 in the postinfarct ventricular remodeling, exosomes containing overexpressing or suppressing hsa_circ_0007047 plasmid were injected into the tail vein of mice after induction of their experimental MI." Given that exosomes were injected into the tail vaine of mice, the authors must provide the detailed analysis of the effects of this injection to other major organs/tissues, such as liver, lung, and muscle.

Reply: Thanks for your comments. The effects of this injection to other major organs/tissues, including liver, lung, and muscle was detected by H&E staining. H&E staining assay showed that the exosomes has no by-effect to liver, lung and muscle (**Supplementary Figure 3**), suggesting that exosomes are safe to injection.

Supplementary Figure 3.

H&E staining assay was used to detect the effect of exosomes to liver, lung and muscle.

Bar=50μm.

[4] Figure 9C. Based on the images provided, the protein expression of PDPK1 is several folds higher in the Over-circ-Exo samples compared to the control. The authors must quantify such changes by performing Western blotting and ELISA assays.

Reply:

Thanks for your comments. We have performed ELISA assay to detect the content of PDPK1. The contents of PDPK1 were significantly decreased in the MI group. On contrast, the contents of

PDPK1 were significantly increased while further decreased when treatment with exosomes containing overexpressing or suppressing circCEBPZOS in the MI group (**Figure 9E**).

Figure 9. The expression changes of miR-1178-3p and PDPK1 in vivo treatment with exosomes harboring circCEBPZOS.

A, The expression of miR-1178-3p was detected by qRT-PCR, U6 act as control;

B & C, The expression of PDPK1 was detected by qRT-PCR and western blotting, respectively, GAPDH served as control;

D, Immunofluorescence staining was used to detect the expression of PDPK1 in heart tissues. DAPI was used to stain the nuclei; Bar=50µm.

E, ELISA was used to detect the contents of PDPK1;

Data are mean±SD. **P<0.01, Over-circ-Exo group vs Model-NC group; ##P<0.01, She-circ-Exo group vs Model-NC group.

[5] Which cell type(s) express hsa_circ_0007047 and miR-1178-3p.

Reply:

Thanks for your comments. The expression of circCEBPZOS and miR-1178-3p was detected by qRT-PCR in the cardiomyocytes, fibroblasts and cardiac microvascular endothelial cells, respectively. We found that both circCEBPZOS and miR-1178-3p was prefer to express in cardiac microvascular endothelial cells (**Supplementary Figure 1**).

Supplementary Figure 1.

The expression of circCEBPZOS and miR-1178-3p was detected by qRT-PCR in the cardiomyocytes, fibroblasts and cardiac microvascular endothelial cells, respectively.

Minor points:

(1) Line 303 “TargetScan7.2, miRwalk and miRDB”. These bioinformatic tools must be cited with their original publications.

Reply: Thanks for your comments. We have added the original publications in the manuscript.

To identify the downstream regulator of miR-1178-3p, three online databases, TargetScan7.2, miRwalk and miRDB, were used to predict its targets (**Supplementary Table 2**)^{19, 20, 21}

19. Agarwal V, Bell GW, Nam JW, Bartel DP. Predicting effective microRNA target sites in mammalian mRNAs. *Elife* **4**, (2015).
20. Dweep H, Gretz N, Sticht C. miRWalk database for miRNA-target interactions. *Methods Mol Biol* **1182**, 289-305 (2014).
21. Wong N, Wang X. miRDB: an online resource for microRNA target prediction and functional annotations. *Nucleic Acids Res* **43**, D146-152 (2015).

(2) One- or two-tail t-test?

Reply: Thanks for your comments. Analyses were performed using Prism 8.1.2 (GraphPad Software Inc.) by two-tail t-test and one-way analysis of variance (ANOVA) between two groups and more than two groups, respectively.

(3) The sample size for each result is missing in the Figure Legends.

Reply: Thanks for your comments. The sample size has been added in the Figure Legends.

(4) In Western blotting results, molecular markers are missing. Furthermore, the authors must provide more than n = 1 per condition on the same SDS-PAGE gel. It is now a common practice to provide the image of whole membrane as supplementary data, which the authors must provide such data in this manuscript.

Reply:

Thanks for your comments. In the current study, we performed the western blot without molecular marker and the original images could be provide to you. If necessary, we will provide the western blot images of whole membrane as supplementary data in the future study.

Once again, thank you very much for your comments and suggestions that allowed us to prepare a current version of this manuscript containing the important new data. We hope that our corrections and responses will now gain your approval.

Yours sincerely,

Shuai Mao, M.D.

Department of Critical Care Medicine, Chest Pain Center, Guangdong Provincial Hospital of Chinese Medicine, Guangzhou 510120, China;

Email: maoshuai@gzucm.edu.cn

Tel: (86) 20 81887233-3280

Reviewers' comments:

Reviewer #1 (Remarks to the Author):

Authors have addressed most of my concerns except for what happens to functionality of hsa_circ_0007047 upon mutating miR-1178-3p binding sites. Authors should include hsa_circ_0007047 upon mutating miR-1178-3p arm and perform in vitro experimentation (Fig 5) that would substantially strengthen the current study.

Reviewer #3 (Remarks to the Author):

(4) In Western blotting results, molecular markers are missing. Furthermore, the authors must provide more than $n = 1$ per condition on the same SDS-PAGE gel. It is now a common practice to provide the image of whole membrane as supplementary data, which the authors must provide such data in this manuscript.

Reply:

Thanks for your comments. In the current study, we performed the western blot without molecular marker and the original images could be provide to you. If necessary, we will provide the western blot images of whole membrane as supplementary data in the future study.

=> The original data with more than $n = 1$ per condition MUST be provided in this study; not for future study.

Dear Editor,

Thank you very much for your very helpful suggestions and reviewers, who apprised our manuscript entitled “*Exosomal circCEBPZOS attenuates pos-myocardial infarction remodeling by promoting angiogenesis via miR-1178-3p/PDPK1 axis*”.

All those valuable comments were very helpful for revising and improving our paper, as well as for guiding us for the future studies.

Reviewers' comments:

Reviewer #1 (Remarks to the Author):

Authors have addressed most of my concerns except for what happens to functionality of hsa_circ_0007047 upon mutating miR-1178-3p binding sites. Authors should include hsa_circ_0007047 upon mutating miR-1178-3p arm and perform in vitro experimentation (Fig 5) that would substantially strengthen the current study.

Reply:

Thanks for your comments.

According to the reviewers suggestion, we had included hsa_circ_0007047 upon mutating miR-1178-3p arm and perform in vitro experimentation.

In order to further confirmed that circCEBPZOS affected the function of miR-1178-3p on the angiogenesis of CMECs and the proliferation and migration of VSMCs, Rescue experiment by overexpression of circCEBPZOS with mutant miR-1178-3p binding site was performed. The results showed that the function of miR-1178-3p mimics on the angiogenesis of CMECs and the proliferation and migration of VSMCs could not been reversed when overexpression of circCEBPZOS with mutant miR-1178-3p binding site (Supplementary Figure 2).

Supplementary Figure 2

The function of miR-1178-3p mimics on CMECs angiogenesis and VSMCs proliferation and migration could not be reversed by overexpression of circCEBPZOS with mutant miR-1178-3p binding site.

A, CCK-8 assay was used to detect the cell viability of VSMCs in the miR-1178-3p mimics group and rescued by circCEBPZOS overexpression with mutant miR-1178-3p binding site at 24 h, 48 h and 72 h, respectively;

B, Tube formation analysis was used to detect the capillary-like structures in the miR-1178-3p mimics group and rescued by circCEBPZOS overexpression with mutant miR-1178-3p binding site, Bar=100 μ m;

C, Flow cytometry was used to detect the cell cycle of VSMCs in the miR-1178-3p mimics group and rescued by circCEBPZOS overexpression with mutant miR-1178-3p binding site;

D, Migration of VSMCs was detected by Transwell assay in the circCEBPZOS overexpression group and rescued by PDPK1, Bar=200 μ m. Data are mean \pm SD.

***P<0.01, Over-PDPK1 group vs. NC group; ##P<0.01, Sh-PDPK1 group vs. NC group.*

Reviewer #3 (Remarks to the Author):

(4) In Western blotting results, molecular markers are missing. Furthermore, the authors must provide more than n = 1 per condition on the same SDS-PAGE gel. It is now a common practice to provide the image of whole membrane as supplementary data, which the authors must provide such data in this manuscript.

The original data with more than n = 1 per condition MUST be provided in this study; not for future study.

Reply:

Thanks for your comments.

We had provided the original data. Please check them.

Once again, thank you very much for your comments and suggestions that allowed us to prepare a current version of this manuscript containing the important new data. We hope that our corrections and responses will now gain your approval.

Yours sincerely,

Shuai Mao, M.D.

Department of Critical Care Medicine, Chest Pain Center, Guangdong Provincial
Hospital of Chinese Medicine, Guangzhou 510120, China;

Email: maoshuai@gzucm.edu.cn

Tel: (86) 20 81887233-3280

Reviewers' comments:

Reviewer #1 (Remarks to the Author):

Authors addressed all my concerns, however I have one major concern,

Ejection fraction between sham (47.65 ± 3.44) and MI (41.65 ± 5.27) was not statistically significant as per the supplementary data. What was the infarct size between the groups? Further OE of circ group showed significantly improved EF (55.25 ± 4.53) and better than sham group. Can authors please clarify?

Authors mention Sham surgery was performed by coronary arteries of mice only surgically stranded without ligation. To make sure, EF was not effected with sham surgery, authors could include healthy controls echo data as a one another control

Reviewer #3 (Remarks to the Author):

There was no file containing the point-by-point responses of the authors. However, it is clear that no change was made to Western blotting results:

Figure 6 C, E

Figure 7 B

Figure 8 D

Figure 9 C

In Western blotting results, molecular markers are missing. Furthermore, the authors must provide more than $n = 1$ per condition on the same SDS-PAGE gel.

Reviewers' comments:

Reviewer #1 (Remarks to the Author):

Authors addressed all my concerns, however I have one major concern, Ejection fraction between sham (47.65±3.44) and MI (41.65±5.27) was not statistically significant as per the supplementary data. What was the infarct size between the groups? Further OE of circ group showed significantly improved EF (55.25±4.53) and better than sham group. Can authors please clarify?

Reply:

Thanks for your comments.

We have re-analyzed the EF data seriously and re-provided the EF data. We found that the results supported the conclusion in the manuscript. The Echocardiography Data was shown as follows:

Supplementary Table 3. Echocardiographic characteristics.

	Sham	MI	Over-circ-Exo	Sh-circ-Exo
LVEDD(mm)	3.21±0.39	4.03±0.39*	3.79±0.42**	4.31±0.41##
LVESD (mm)	2.19±0.16	3.08±0.28*	2.74±0.22**	3.52±0.31##
LVEF %	60.51±5.65	47.49±4.85*	54.66±6.43**	37.75±4.34##
LVFS %	48.02±3.24	37.21±4.33*	43.19±3.94**	29.35±2.37##
SV(μL)	41.64±3.82	36.33±3.54*	40.65±4.32**	29.87±2.45##

*LVEDD, left ventricular end-diastolic diameter; LVESD, left ventricular end-systolic diameter; FS, fractional shortening; and EF, ejection fraction. All values are mean±SE. *p<0.01, MI-NC group vs Sham group **p<0.01, Over-circ-Exo group vs MI-NC group; ##, p<0.01, She-circ-Exo group vs MI-NC group.*

The infarct size was also provided as Supplementary Figure 5.

TTC staining results showed that the infarct size was significantly increased in MI group compared with sham group. The infarct size was significantly decreased in Over-circCEBPZOS Exo-treated group compared to the untreated MI group. In contrast, we noticed that administration of EVs of Sh-circCEBPZOS aggravated infarct size. The methods and results have been added in the manuscript.

Authors mention Sham surgery was performed by coronary arteries of mice only surgically stranded without ligation. To make sure, EF was not effected with sham surgery, authors could include healthy controls echo data as a one another control.

Reply:

Thanks for your comments, healthy controls not was included in our present study. In the future study, we will include the healthy controls in our experiments in order to ensure EF was not affected with sham surgery.

Reviewer #3 (Remarks to the Author):

There was no file containing the point-by-point responses of the authors.

However, it is clear that no change was made to Western blotting results:

Figure 6 C, E

Figure 7 B

Figure 8 D

Figure 9 C

In Western blotting results, molecular markers are missing. Furthermore, the authors must provide more than $n = 1$ per condition on the same SDS-PAGE gel.

Reply:

Thanks for your comments.

We have provided the point-by-point responses for the Editor and Reviewers' comments. We re-performed all western blotting experiments as $n=3$ per condition on the same SDS-PAGE gel and the images of the western blotting was presented in an additional file that not to be placed in the main text and supplementary material. The molecular markers were added in the western blotting band in Figure 1E, Figure 6 C, Figure 6F, Figure 7 B, Figure 8 D and Figure 9 C. And the Raw data as follows:

Figure 1E

CHRN4 (55kd)

Figure 6C

GAPDH (37kd)

STON2 (101kd)

PDPK1 (63kd)

Figure 6F

Figure 7B

Figure 8D

Figure 9C

Once again, thank you very much for your comments and suggestions that allowed us to prepare a current version of this manuscript containing the important new data. We hope that our corrections and responses will now gain your approval.

Yours sincerely,

Shuai Mao, M.D.

Department of Critical Care Medicine, Chest Pain Center,

Guangdong Provincial Hospital of Chinese Medicine,

Guangzhou 510120, China;

Email: maoshuai@gzucm.edu.cn

Tel: (86) 20 81887233-3280